# The ubiquitin ligase RNF5 determines acute myeloid leukemia growth and susceptibility to histone deacetylase inhibitors

Ali Khateb[1,2], Anagha Deshpande[2], Yongmei Feng[2], Darren Finlay [2], Joo Sang Lee [3], Ikrame Lazar[1,2], Bertrand Fabre[1,2], Yan Li[2], Yu Fujita [2,9], Tongwu Zhang [4], Jun Yin[2], Ian Pass[2], Ido Livneh[1], Irmela Jeremias [5], Carol Burian[6], James R. Mason[6], Ronit Almog[7], Nurit Horesh[8], Yishai Ofran[1,8], Kevin Brown [4], Kristiina Vuori [2], Michael Jackson[2], Eytan Ruppin[3], Aniruddha J. Deshpande [2] & Ze'ev A. Ronai [2✉]

Acute myeloid leukemia (AML) remains incurable, largely due to its resistance to conventional treatments. Here, we find that increased abundance of the ubiquitin ligase RNF5 contributes to AML development and survival. High RNF5 expression in AML patient specimens correlates with poor prognosis. RNF5 inhibition decreases AML cell growth in culture, in patient-derived xenograft (PDX) samples and in vivo, and delays development of MLL-AF9–driven leukemogenesis in mice, prolonging their survival. RNF5 inhibition causes transcriptional changes that overlap with those seen upon histone deacetylase (HDAC)1 inhibition. RNF5 induces the formation of K29 ubiquitin chains on the histone-binding protein RBBP4, promoting its recruitment to and subsequent epigenetic regulation of genes involved in AML maintenance. Correspondingly, RNF5 or RBBP4 knockdown enhances AML cell sensitivity to HDAC inhibitors. Notably, low expression of both *RNF5* and *HDAC* coincides with a favorable prognosis. Our studies identify an ERAD-independent role for RNF5, demonstrating that its control of RBBP4 constitutes an epigenetic pathway that drives AML, and highlight RNF5/RBBP4 as markers useful to stratify patients for treatment with HDAC inhibitors.

[1] Technion Integrated Cancer Center, Faculty of Medicine, Technion Israel Institute of Technology, Haifa, Israel. [2] Cancer Center, Sanford Burnham Prebys Medical Discovery Institute, La Jolla, CA, USA. [3] Cancer Data Science Lab (CDSL), National Cancer Institute, National Institute of Health, Bethesda, MD, USA. [4] Laboratory of Translational Genomics, Division of Cancer Epidemiology and Genetics, National Cancer Institute, Bethesda, MD, USA. [5] Research Unit Apoptosis in Hematopoietic Stem Cells, Helmholtz Center Munich, German Center for Environmental Health, Munich, Germany. [6] Scripps MD Anderson Cancer Center, La Jolla, CA, USA. [7] Rambam Health Care Campus, Epidemiology Department and Biobank, Haifa, Israel. [8] Rambam Health Care Campus, Hematology and Bone marrow Transplantation Department, Haifa, Israel. [9] Present address: Division of Respiratory Medicine, Department of Internal Medicine, Jikei University School of Medicine, Tokyo, Japan. ✉email: ronai@sbpdiscovery.org

AML is a heterogeneous hematological cancer characterized by the accumulation of somatic mutations in immature myeloid progenitor cells. Such mutations alter the self-renewal, proliferation, and differentiation capabilities of these progenitor cells[1,2]. The prognosis of AML patients is strongly influenced by the type of chromosomal or genetic alterations and by changes in gene expression[1,3]. Although numerous mutations and chromosomal aberrations that drive AML development have been identified[1,4], the molecular components and epigenetic modulators that contribute to AML etiology and pathophysiology are not well defined. Approximately one-third of AML patients fail to achieve complete remission in response to chemotherapy, and 40–70% of those who do enter remission relapse within 5 years. Thus, there is an urgent need to better understand the molecular mechanisms underlying AML development and progression to facilitate the development of more effective therapies.

RING finger protein 5 (RNF5) is an ER-associated E3 ubiquitin ligase and component of the UBC6e-p97 complex, which is implicated in ER-associated degradation (ERAD)[5,6], a pathway involved in maintaining protein homeostasis. RNF5 recognizes misfolded proteins and promotes their ubiquitination and proteasome-dependent degradation[5,6]. *RNF5* expression is relatively increased in several cancers, including breast cancer, hepatocellular carcinoma, and AML[7]. RNF5 regulates glutamine metabolism by promoting degradation of misfolded glutamine carrier proteins, a function important in the cancer cell response to ER stress-inducing chemotherapies such as paclitaxel[8]. RNF5 promotes degradation of the protease ATG4B, which limits basal amounts of autophagy[9]. RNF5 also limits intestinal inflammation by controlling S100A8 protein stability[10]. Given the important pathophysiological roles of RNF5, the observation that it is upregulated in AML cells and patient samples prompted us to investigate the possible contribution of RNF5 to the development and progression of this disease.

Histone modification by acetylation contributes to the dynamic regulation of chromatin structure and affects gene expression programs. Histone acetylation status is associated with transcriptional regulation of leukemic fusion proteins, such as AML1-ETO, PML-RARα, and MLL-CBP[11,12]. Correspondingly, histone deacetylases (HDACs) are implicated in the etiology and progression of leukemia[13], and HDAC inhibitors can block growth or promote differentiation or apoptosis of leukemia cells[14]. The retinoblastoma binding protein 4 (RBBP4) is a component of multi-protein complexes that function in nucleosome assembly and histone modifications, modulate gene transcription, and regulate the cell cycle and proliferation[15]. Such complexes include the nucleosome remodeling and deacetylase (NuRD) complex, polycomb repressor complex 2 (PRC2), and switch independent 3 A (SIN3A)[15,16]. Overexpression of *RBBP4* or *HDAC1* correlates with clinicopathologic characteristics and poor prognosis in breast cancer[17], and *RBBP4* expression positively correlates with hepatic metastasis and poor prognosis in colon cancer patients[18]. RBBP4 is also implicated in the regulation of DNA repair genes, and its suppression in glioblastoma enhances tumor sensitivity to temozolomide chemotherapy[19]. However, the function of RBBP4 in AML has not been studied.

Here, we identified a central role for the RNF5-RBBP4 axis in AML maintenance and responsiveness to HDAC inhibitors. Our data suggest that targeting RNF5 and HDAC pathways represents a therapeutic modality for AML and that RNF5 or RBBP4 abundance could serve as a prognostic marker and means to stratify patients for treatment with HDAC inhibitors.

## Results

### Increased abundance of RNF5 in AML patient samples correlates with poor prognosis.
Analysis of RNA-seq datasets for various cancer cells in the Cancer Cell Line Encyclopedia (CCLE) database identified a higher copy number and levels of *RNF5* transcripts in AML, chronic myeloid leukemia (CML), and T-cell acute lymphoblastic leukemia (T-ALL) relative to other tumor types (Fig. 1a and Supplementary Fig. 1a). Higher levels of RNF5 protein were confirmed in AML and CML cell lines compared with other cancer lines (Supplementary Fig. 1b). To assess the clinical relevance of RNF5 expression in AML, we analyzed levels of RNF5 mRNA and protein in peripheral blood mononuclear cells (PBMCs) from independent cohorts of AML patients. Similar to results in AML lines, the average abundance of RNF5 protein was significantly higher in PBMCs from AML patients relative to control samples (CD34$^+$ and PBMCs) (Fig. 1b, c, Supplementary Fig. 1c, d). The patient cohort included an equal number of females (24; median age = 59) and males (24; median age = 64.5; Supplementary Table 1). Given that RNF5 is a ubiquitin ligase, its transcript levels are not as reflective of activity as are protein levels, as self-degradation or other post-translational modifications can alter RNF5 subcellular localization, availability and/or activity. Indeed, analysis of patient cohorts revealed a significant increase in RNF5 protein but not transcript levels in patients as compared to healthy subjects (Fig. 1b, c and Supplementary Fig. 1e). Stratification of the 50 patients into two groups based on the top high ($n = 8$, 15%) and low ($n = 42$, 85%) RNF5 protein levels revealed that high abundance coincided with poor overall survival ($P = 0.05$, Fig. 1d). Notably, this difference was not due to blast counts, as they did not differ significantly among patients showing high or low RNF5 levels (Supplementary Fig. 1f, g). Independent analysis of AML patients ($n = 154$) based on The Cancer Genome Atlas (TCGA) dataset confirmed a significant positive correlation between high *RNF5* expression (10%) and poor survival ($P = 0.009$, Fig. 1e). Notably, there was no pattern of RNF5 abundance that either positively or negatively correlated with the presence of *FLT3* or *NPM1* mutations (Supplementary Fig. 1h, i), suggesting that the significance of RNF5 activity to AML may not depend on any specific oncogenic driver(s) or activation of particular signaling pathways.

Assessment of an independent AML patient cohort (from the Rambam Health Campus Center, Haifa, Israel which included multiple samples obtained from 5 females with a median age of 62 and 6 males with a median age of 63.5, as detailed in Supplementary Table 1) confirmed higher levels of RNF5 protein in AML patient blood samples ($n = 18$) relative to samples taken from healthy donors ($n = 5$) (Fig. 1f, g). Because this cohort included samples taken from patients both prior to and following therapy, we were able to compare RNF5 abundance before and after therapy and at remission or relapse stages. Notably, RNF5 abundance markedly decreased following chemotherapy and during remission ($n = 8$) (Fig. 1h, i and Supplementary Fig. 1j). Conversely, RNF5 levels at diagnosis were similar to those seen in patients that either relapsed or were refractory to treatment ($n = 5$) (Fig. 1h, j). These results suggest that RNF5 levels in AML blasts may serve as a prognostic marker for AML.

### RNF5 is required for AML cell proliferation and survival.
Next, we asked how RNF5 knockdown (RNF5-KD) would impact leukemia cell growth in vitro. Interestingly, KD using RNF5-targeting short hairpin RNAs (shRNF5) decreased viability and attenuated growth of MOLM-13 and U937 AML lines (Fig. 2a and Supplementary Fig. 2a) but not of CML (K-562) or T-ALL (Jurkat) lines (Supplementary Fig. 2b, c). RNF5 KD in MOLM-13 or U937 AML cells also promoted accumulation of cells in the G1 phase of the cell cycle (Fig. 2b), an effect accompanied by increase in levels of the cell cycle regulatory proteins p27 and p21 (Fig. 2c). Moreover, AML cells MOLM-13 and U937 with RNF5-KD

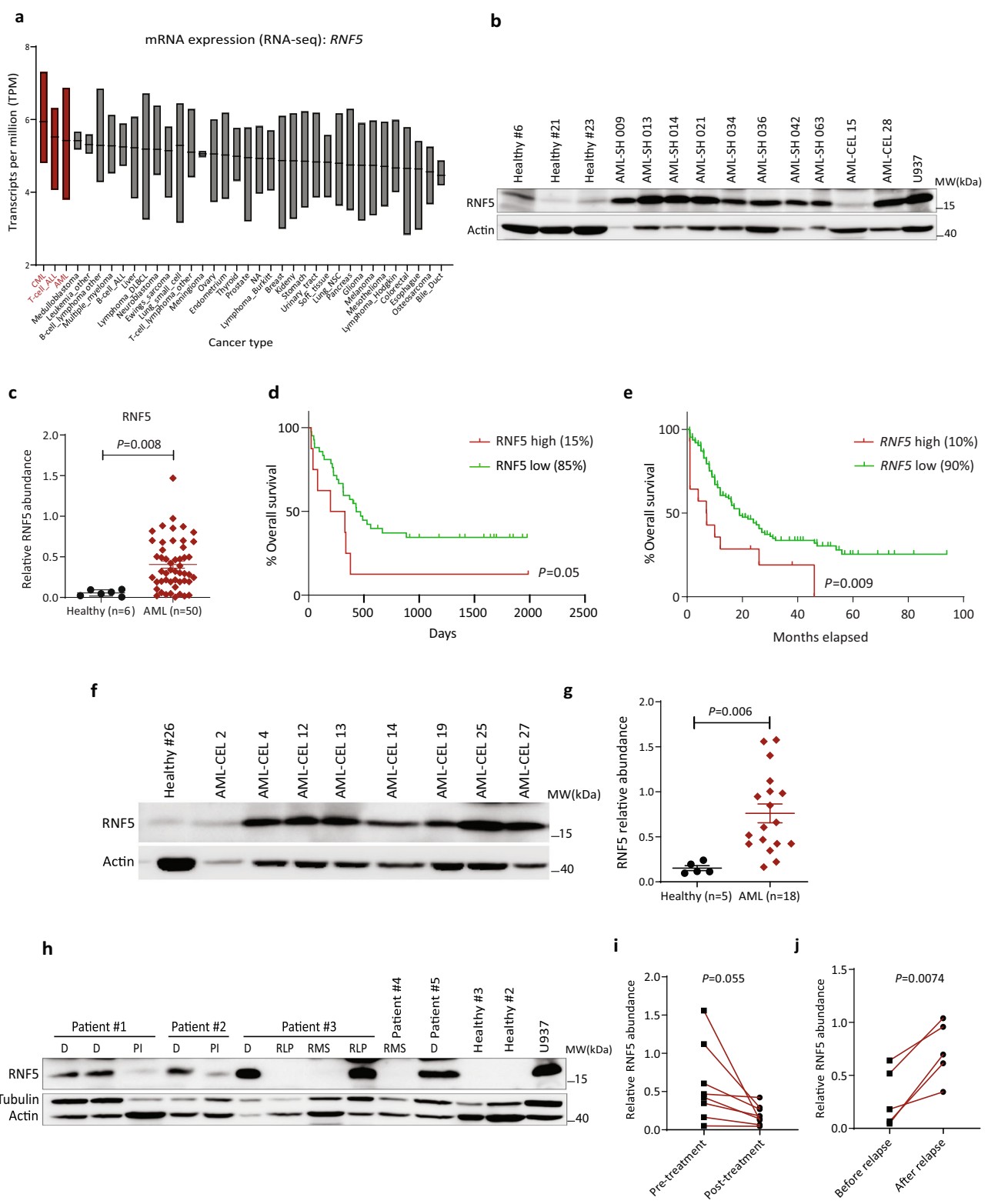

showed reduced colony formation in soft agar relative to controls (Fig. 2d) and increased abundance of proteins associated with apoptosis, as reflected by levels of cleaved forms of caspase-3 (Fig. 2e) and poly ADP ribose polymerase (PARP) (Supplementary Fig. 2d). Effects of RNF5-KD on U937 and MOLM-13 cells were confirmed in two additional AML cell lines HL-60 and THP-1 (Supplementary Fig. 2e–h). Importantly, re-expression of

RNF5 WT, but not the catalytically inactive RING mutant (RNF5 RM), restored cell proliferation (Fig. 2f, g), confirming the specificity of these phenotypes and suggesting that RNF5 catalytic activity is required for AML cell proliferation.

To verify changes seen upon KD, we used CRISPR-Cas9 gene-editing technology to deplete RNF5 in MOLM-13 cells stably expressing Cas9 using RNF5-targeting guide RNAs (sgRNAs).

**Fig. 1 RNF5 expression in AML cell lines and patient samples. a** *RNF5* expression data obtained from the CCLE RNA-seq datasets. Transcripts per million (TPM) for protein-coding genes were calculated using RSEM software. Data is log2-transformed, using a pseudo-count of 1w. Box plot is sorted and colored by the average distribution of *RNF5* expression in a lineage. Lineages are composed of a number of cell lines. The highest average distributions are shown at left in red. The line within a box represents the mean. **b** Representative western blot analysis of RNF5 in peripheral blood mononuclear cells (PBMCs) from healthy donors and AML patients (Scripps Health). **c** Relative abundance of RNF5 protein in PBMCs from AML patients ($n = 50$) and healthy control subjects ($n = 6$) from the Scripps Health Center. RNF5 abundance in U937 cells served as a reference for quantification (presented as the mean ± SEM). $P = 0.008$ by unpaired two-tailed *t*-test. **d** Kaplan–Meier survival curve analysis of AML patients stratified by top high ($n = 8$) versus low ($n = 42$) RNF5 protein (Scripps Health). $P = 0.05$ by two-sided Mantel–Cox log-rank test. **e** Kaplan–Meier survival curve of AML patients stratified by top high ($n = 14$) versus low ($n = 150$) *RNF5* transcript levels (TCGA dataset). $P = 0.009$ by tow-sided Mantel–Cox log-rank test. **f, h** Levels of RNF5 protein and actin in PBMCs from healthy controls and AML patients (Rambam Health Campus Center). D diagnosis, PI post induction, RLP relapse, RMS remission. Quantified data are presented as the mean ± SEM. $P = 0.006$ by unpaired two-tailed *t*-test. **g** Relative abundance of RNF5 protein in PBMCs from AML patients ($n = 18$) and healthy controls ($n = 5$) from the Rambam Center. Quantified data are presented as the mean ± SEM. $P = 0.006$ by unpaired two-tailed *t*-test. **i, j** Relative abundance of RNF5 protein in AML patient samples (PBMCs) collected before and after induction treatment (**i**, $n = 8$) or before and after relapse (**j**, $n = 5$). Lines connect values for the same patient. $P = 0.055$, $P = 0.0074$ by paired two-tailed *t*-test. Source data are provided as a Source Data file.

Relative to control cells transduced with *Renilla* luciferase-targeting sgRNAs, cells transduced with RNF5-targeting sgRNAs showed impaired growth based on CellTiter-Glo luminescence assay (Fig. 2h, i).

To further assess RNF5 function in AML, we monitored the viability of xenografted patient-derived AML cells (PDX, AML-669[20]) transduced with shRNF5 or control constructs. As expected, using two independent shRNF5s (albeit limited KD efficiency), we observed decreased viability of xenografted RNF5-KD cells relative to controls (Fig. 2j–i). These findings confirm our observations in AML cell lines and support the notion that RNF5 downregulation impairs the proliferation of AML blasts.

**RNF5 inhibition enhances ER stress-induced apoptosis of AML cells**. As RNF5 functions as part of ERAD and the ER stress response, we asked whether changing RNF5 abundance would alter the ER stress response in AML cells. To do so, we exposed RNF5-KD or control MOLM-13 cells to thapsigargin or tunicamycin to inhibit the ER $Ca^{2+}$-ATPase (SERCA) or protein glycosylation, respectively[21], as a mean to induce ER stress. Thapsigargin treatment of MOLM-13 RNF5-KD cells increased apoptotic markers to levels higher than those seen in control cells (Fig. 3a, b). Thapsigargin treatment also decreased the viability of MOLM-13 RNF5-KD cells to a greater extent than seen in control MOLM-13 WT RNF5 cells (Fig. 3c). Tunicamycin treatment also decreased the viability of RNF5-KD HL-60 cells compared to tunicamycin-treated control HL-60 cells (Supplementary Fig. 3a). Consistent with a function in ER stress, RNF5-KD increased levels of transcripts encoding key UPR components, including *CHOP*, *ATF3*, and *sXBP1*, in thapsigargin-treated MOLM-13 (Fig. 3d) and HL-60 cells (Supplementary Fig. 3b), relative to mock-transduced controls.

Given the link between ER stress and proteasomal degradation, we assessed potential synergy between RNF5 KD and proteasomal inhibition. Indeed, RNF5-KD MOLM-13 cells treated with the proteasome inhibitor bortezomib (BTZ) showed increased levels of apoptotic markers such as cleaved forms of caspase-3 and PARP (Fig. 3e) and decreased viability (Fig. 3f) relative to control-treated cells. Using annexin V and propidium iodide staining, which monitor the degree of programmed cell death, we showed that RNF5-KD also enhanced apoptosis of BTZ-treated HL-60 cells (Fig. 3f), decreasing the BTZ IC$_{50}$ from 9.6 nM in controls to 5.4 nM in RNF5-KD cells (Fig. 3g). These data suggest that RNF5 plays a role in the response of AML cells to proteotoxic stress.

**RNF5 loss delays leukemia establishment and progression**. We next asked whether RNF5 activity modulates leukemia growth in vivo. To do so, we used a human AML xenograft model in which luciferase-expressing U937 cells (U937-pGFL) were transduced with doxycycline-inducible shRNF5 or control shRNA before being injected intravenously into NOD/SCID mice (Supplementary Fig. 4a). Following leukemia establishment, as confirmed by bioluminescence, mice were fed a doxycycline-containing diet and monitored for disease progression and overall survival. Interestingly, animals injected with RNF5-KD cells exhibited a markedly decreased leukemia burden and prolonged survival relative to control mice (Fig. 4a, b and Supplementary Fig. 4b). RT-PCR analysis of splenocytes isolated from mice transplanted with RNF5-KD cells confirmed expression of shRNF5 (Supplementary Fig. 4c). Western blot analysis of splenocyte lysates revealed more abundant expression of the cell cycle regulatory protein p27 in shRNF5 relative to control cells (Supplementary Fig. 4d), consistent with our in vitro data and with the delayed leukemia progression observed following RNF5 KD (Fig. 4a, b). Collectively, these data indicate that RNF5 is required for AML cell proliferation in vivo.

We also asked whether RNF5 functions in AML initiation using the MLL-AF9 model[22] for in vitro and in vivo studies. For in vitro analysis, we purified hematopoietic stem and progenitor (Lin-depleted) cells (HSPCs) from the bone marrow of *Rnf5*$^{-/-}$, which exhibit normal development and hematopoiesis[23], and wild-type (WT) C57/BL6 mice. HSPCs from these mice were retrovirally transduced with a bicistronic construct harboring MLL-AF9 linked to a green fluorescent protein (GFP) marker. When we assessed colony-forming capacity (CFC), we found that compared to WT GFP-MLL-AF9 cells, *Rnf5*$^{-/-}$ GFP-MLL-AF9 cells exhibited markedly reduced CFC in methylcellulose after 7, 14, and 21 days in culture and observed a striking reduction in the number of blast-like colonies (Fig. 4d, e). These phenotypes are consistent with apparent terminal differentiation of *Rnf5*$^{-/-}$ cells, as reflected by a greater cytoplasm/nucleus ratio and more vacuolated cytoplasm (Fig. 4e, f).

To assess leukemogenesis in vivo, we injected sublethally-irradiated WT C57/BL6 recipient mice with GFP-MLL-AF9–transduced *Rnf5*$^{WT}$ or *Rnf5*$^{-/-}$ cells and monitored cell engraftment by flow cytometry for GFP-positive (GFP+) cells in peripheral blood (Fig. 4c). Analysis on days 15 and 28 post-injection identified fewer GFP+ cells in mice injected with GFP-MLL-AF9 *Rnf5*$^{-/-}$ cells than in mice injected with GFP-MLL-AF9 *Rnf5*$^{WT}$ cells, indicating a delay in leukemia development (Fig. 4g). Moreover, mice harboring GFP-MLL-AF9 *Rnf5*$^{-/-}$ cells exhibited prolonged survival relative to mice injected with GFP-MLL-AF9 *Rnf5*$^{WT}$ cells (Fig. 4h). Collectively, these data show that RNF5 loss decreases the colony-forming capacity of MLL-AF9–transformed pre-leukemic cells in vitro and delays leukemia progression in vivo.

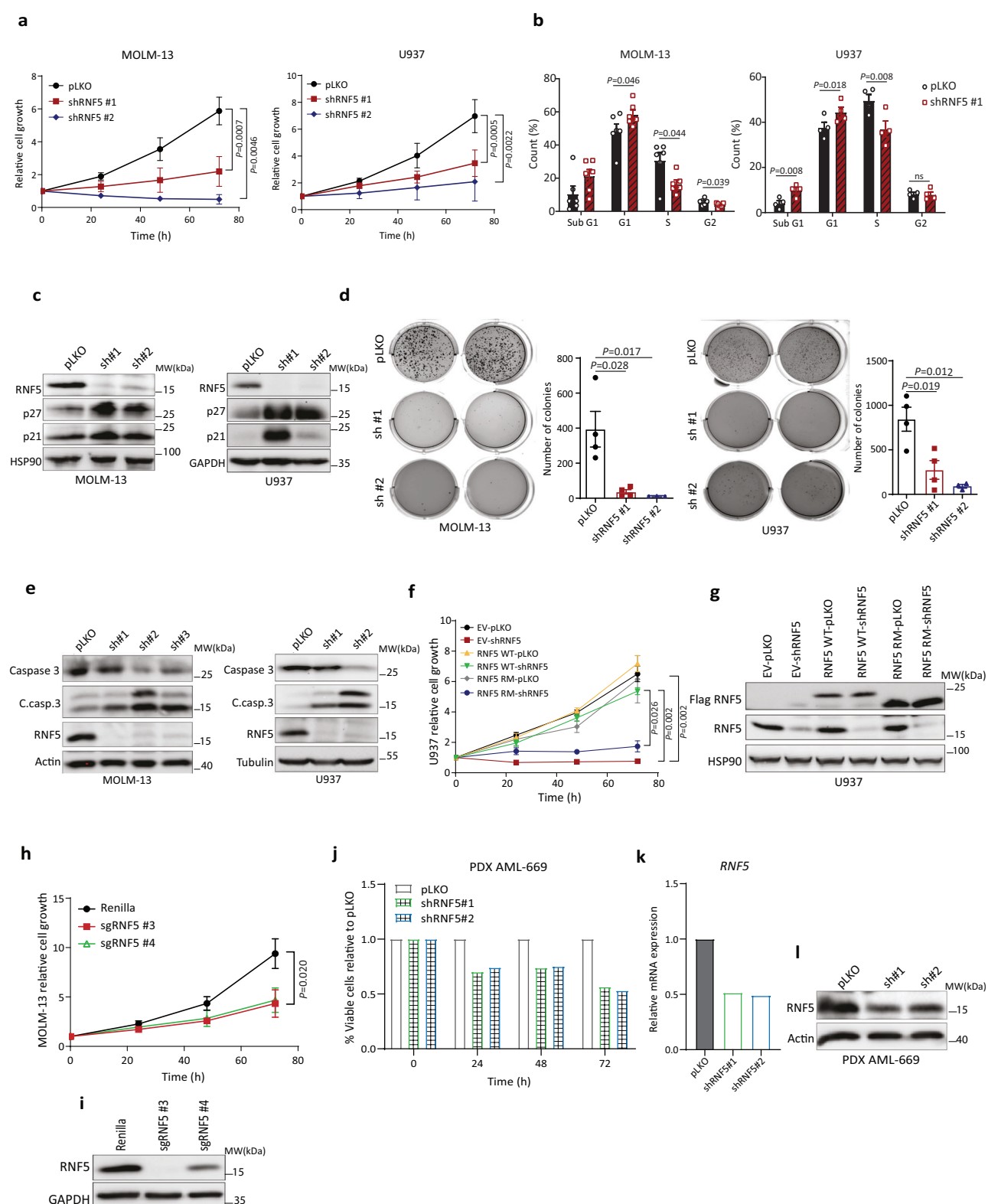

**RNF5 activity modulates transcription in AML cells**. To identify pathways modulated by RNF5 activity in AML cells, we first monitored transcriptional changes in MOLM-13, U937, and HL-60 AML lines expressing either RNF5-KD or control constructs. RNA sequencing (RNA-seq) analysis identified a total of 237, 814, and 1380 dysregulated genes in MOLM-13, U937, and HL-60, respectively, following RNF5 KD relative to control (RNF5-WT)

cells (Fig. 5a, b and Supplementary Data file 1). Ingenuity Pathway Analysis identified selective enrichment of genes implicated in myeloid cell function such as NF-κB signaling, IL-8 signaling, reactive oxygen species, and several pathways related to cell migration such as Rho GTPase and Tec kinase signaling (Supplementary Fig. 5a). Expression of a total of 59 genes (35 up- and 24 downregulated) were significantly altered by RN5F KD in all

**Fig. 2 RNF5 is required for AML cell proliferation and survival. a** Growth assay of MOLM-13 and U937 cells transduced with empty vector (pLKO) or two different shRNF5 constructs. Cell growth was analyzed using CellTiter-Glo assay. **b** Cell cycle analysis of MOLM-13 and U937 cell lines 5 days after transduction. **c** Western blot analysis of cell cycle regulatory proteins in MOLM-13 and U937 AML cells 5 days after transduction. **d** Representative images (left) and quantification (right) of colonies in soft agar from MOLM-13 and U937 were assessed after 15 days in culture. **e** Western blot analysis of apoptosis-related proteins in indicated lines 5 days after transduction (C.casp.3 = cleaved caspase-3). **f** Growth assay of U937 cells expressing doxycycline-inducible Flag-tagged RNF5 WT, RM or empty vector (EV). Cells were induced 48 h with doxycycline (1 μg/ml) and then transduced with either empty vector (pLKO) or shRNF5 for 5 days. **g** Western blot analysis of indicated proteins in U937 transduced as described in (**f**). **h** Growth assay of MOLM-13 cells stably expressing Cas9 and transduced with control *Renilla*-targeting sgRNA or two RNF5-targeting sgRNAs. CRISPR was performed based on CRISPR knockout cell pool. **i** Western blot analysis of indicated proteins in MOLM-13 cells described in (**h**). **j** Growth assay of PDX AML-669 cells after transduction with empty vector (pLKO) or two independent shRNF5 constructs. Quantified data are presented as the mean ± SD of two independent experiments. (**k, l**) RT-qPCR (**k**) and western blot (**l**) analyses confirming RNF5 KD in PDX AML-669 transduced as described in (**j**). Quantified data are presented as the mean ± SD (**a, f,** and **h**) or SEM (**b** and **d**) of $n = 5$ (**a** (left)), $n = 6$ (**a** (right), **b** (left)), or $n = 4$ (**b** (right), **d, f,** and **h**) independent experiments. Western blot data are representative of three experiments. *P* values were determined using two-way ANOVA followed by Tukey's multiple comparison test (**a, f,** and **h**) or paired two-tailed *t*-test (**b,** and **d**). Source data are provided as a Source Data file.

three AML lines (Fig. 5a, b). Among upregulated genes were *CDKN1A* and *CDKN2D*, which encode cell cycle inhibitors; *LIMK1*, which encodes a kinase functioning in regulation of the actin cytoskeleton; *ANXA1*, which encodes a calcium-binding protein functioning in metabolism, EGFR signaling, and cell death programs; and *NCF1*, which encodes a subunit of NADPH oxidase (Fig. 5c and Supplementary Fig. 5b). Downregulated genes included antiapoptotic *BCL2A1*, and *SAP18*, which encodes a histone deacetylase complex subunit functioning in transcriptional repression (Supplementary Fig. 5b). Moreover, such changes were consistent with phenotypic changes seen in RNF5-KD AML cell lines, such as reduced proliferation and increased apoptosis. Interestingly, analysis of the Library of Integrated Network-Based Cellular Signatures (LINCS) drug screening database identified a notable overlap between transcriptomic changes induced by the HDAC1 inhibitor mocetinostat in various cancer cells and those seen in shRNF5-expressing MOLM-13 and HL-60 cells (Fig. 5d). Five and one out of the top ten transcriptional changes identified in LINCS following HDAC inhibition overlapped with those seen following RNF5 KD in MOLM-13 and HL-60 cells, respectively (Fig. 5d, Supplementary Fig. 5c and Supplementary Data file 2). Among commonly affected pathways were activation of GP6 and Rho GTPase signaling and repression of the nucleotide excision repair (NER) pathway (Fig. 5e). These observations suggest that RNF5 may regulate HDAC activity in AML cells.

**RNF5 interacts with and ubiquitinates the retinoblastoma binding protein 4.** We hypothesized that RNF5 elicits transcriptional changes through intermediate regulatory component (s). Thus, to identify RNF5-interacting proteins or substrates we performed liquid chromatography-tandem mass spectrometry (LC-MS/MS) and compared proteins immunoprecipitated from lysates of MOLM-13 cells expressing inducible Flag-tagged RNF5 versus those expressing empty vector. Among 65 RNF5-interacting proteins identified were previously reported substrates, such as 26 S proteasome components, VCP and S100A8[5,10], as well as proteins implicated in AML development, such as DHX15[24] and gelsolin[25,26] (Supplementary Data file 3). Among the more abundant RNF5-bound proteins were components of ERAD, translation initiation, proteolytic and mRNA catabolic processes (Fig. 5f, Supplementary Fig. 5d, e and Supplementary Data file 3).

Although none of the interacting proteins identified here were transcription factors, epigenetic modifications initiated by changes in RNF5 expression could also underlie changes in gene expression. In fact, we identified one RNF5-interacting protein as the epigenetic regulator histone-binding protein RBBP4 (Fig. 5f and Supplementary Data file 3). Analysis of transcriptome data

from TCGA revealed an inverse correlation of *RBBP4* expression with expression of genes upregulated in RNF5-KD cells (Fig. 5g and Supplementary Fig. 5f), suggesting that RNF5 positively controls RBBP4 transcriptional regulatory function. RBBP4 is a component of several chromatin assembly, remodeling, and nucleosome modification complexes, including PRC2[27] and the NuRD co-repressor complex, which contains HDAC1 and HDAC2[28]. Indeed, the inverse correlation between *RBBP4* expression and RNF5-upregulated genes was mirrored when we analyzed *HDAC1, HDAC2,* and *EZH2* expression relative to RNF5-upregulated genes (Supplementary Fig. 5g–i). Increased *RBBP4* expression was also correlated with malignant phenotypes of several human tumors including AML[29]. Analysis of tumor data in TCGA revealed high *RBBP4* expression in AML, compared with other tumor types (Supplementary Fig. 5j). Assessment of an AML patient cohort confirmed higher RBBP4 expression in samples from AML patients compared to healthy donors (Fig. 5h and Supplementary Fig. 5k). Stratification of AML patients based on *RBBP4* expression indicated that high expression (the top 30%) correlated with poor overall survival (Fig. 5i).

If RNF5 positively regulates RBBP4, RBBP4 KD should promote phenotypic changes in AML cells similar to RNF5 KD. Indeed, shRNA-based RBBP4 KD in MOLM-13 and U937 cells impaired their growth (Fig. 5j and Supplementary Fig. 5l), promoted PARP cleavage indicative of apoptosis (Fig. 5k and Supplementary Fig. 5m), and induced expression of genes that were also induced by RNF5-KD (Fig. 5l). Furthermore, when we examined RBBP4 function in vivo using the U937 xenograft model, mice harboring RBBP4 KD in xenografted cells showed delayed AML development and prolonged survival relative to WT-RBBP4 controls, phenocopying changes seen upon RNF5 KD (Fig. 5m, n and Supplementary Fig. 5n, o). Notably, western blot analysis of splenocytes from mice transplanted with RBBP4 KD cells showed that these cells retained RBBP4 expression (Supplementary Fig. 5p). Since RBBP4 KD was confirmed prior to injection of these cells, it is likely that these cells emerged as escapers during in vivo selection (Supplementary Fig. 5n, o). The latter explains the shorter survival observed in mice that harbored the escapers, compared with mice that retained RBBP4 KD (Fig. 5n). Interestingly, similar to outcomes seen in RNF5-KD AML cells, RBBP4 KD blocked the growth of AML, but not CML and T-ALL, cell lines (Supplementary Fig. 6a, b), confirming a link between RNF5 and RBBP4 in the context of AML.

RNF5 is a transmembrane protein primarily associated with the ER, and its ubiquitin ligase domain is located in the cytosol[6,30]. We assessed potential interaction between RNF5 and RBBP4 in the HEK293T line by coimmunoprecipitation of ectopically expressed WT RNF5, a catalytically inactive RING mutant (RNF5 RM), or a

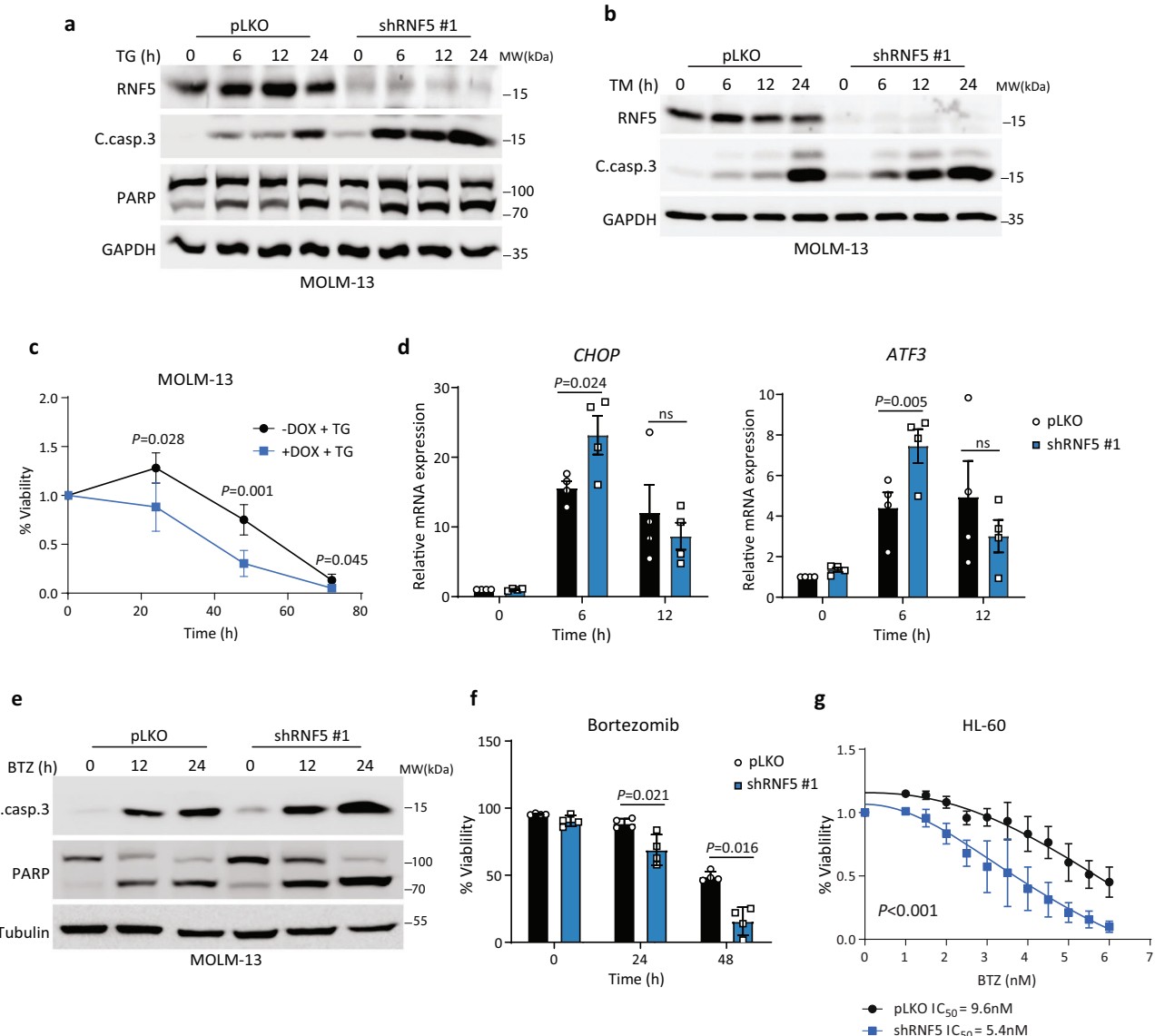

**Fig. 3 RNF5 inhibition sensitizes AML cells to ER stress-induced apoptosis. a, b** Western blot analysis of indicated proteins in MOLM-13 cells expressing empty vector (pLKO) or shRNF5 #1 and treated with thapsigargin (TG, 1 μM) (**a**) or tunicamycin (TM, 2 μg/ml) (**b**) for indicated times. **c** Luminescence-based viability assay of MOLM-13 cells expressing inducible shRNF5 and treated with or without doxycycline (DOX, 1 μg/ml) for 3 days before treatment with TG (100 nM) for indicated times. **d** RT-qPCR analysis of *CHOP* and *ATF3* mRNA in MOLM-13 cells expressing pLKO or shRNF5 #1 and treated with TG (100 nM) for indicated times. **e** Western blot analysis of cleaved caspase-3 (C.casp.3) and PARP in MOLM-13 cells expressing pLKO or shRNF5 #1 and treated with bortezomib (BTZ, 5 nM) for indicated times. **f** Fluorescence-based viability assay of HL-60 cells expressing pLKO or shRNF5 and treated with BTZ (5 nM) for indicated times. Cell viability was determined by flow cytometry of cells stained with annexin V conjugated to fluorescein isothiocyanate and propidium iodide. **g** Luminescence viability assay of HL-60 cells expressing pLKO or shRNF5 and treated 48 h with indicated concentrations of BTZ. Data are presented as the mean ± SD (**c**, **f**, and **g**) or SEM (**d**) of $n = 3$ (**c** and **g**), $n = 5$ (**d** (left)), or $n = 4$ (**d** (right) and **f**) independent experiments. *P* values were determined using paired two-tailed *t*-test (**c**, **d**, and **f**) or two-way ANOVA (**g**). ns: not significant. Source data are provided as a Source Data file.

C-terminal transmembrane domain deletion mutant (RNF5 ΔCT) (Fig. 6a). Endogenous RBBP4 coimmunoprecipitated with all RNF5 constructs, suggesting that both the RING and transmembrane domains are dispensable for protein-protein interaction (Fig. 6b). Reciprocal IP using RBBP4 as bait confirmed interaction with RNF5 (Fig. 6c). Interaction between endogenous RBBP4 and ectopically expressed RNF5 was also confirmed in MOLM-13 cells (Fig. 6d). Next, we assessed potential effect of RNF5 on RBBP4 ubiquitination. Co-expression of HA-tagged ubiquitin, Myc-tagged RBBP4, and Flag-tagged RNF5 constructs in HEK293T cells revealed that RBBP4 was ubiquitinated by WT RNF5, but not by RNF5 RM or RNF5 ΔCT (Fig. 6e), indicating that ubiquitin ligase activity (RING

domain-dependent) and membrane association are both required for an RNF5-mediated increase in RBBP4 ubiquitination. Correspondingly, RNF5-KD in HEK293T or MOLM-13 cells decreased RBBP4 ubiquitination relative to controls (Supplementary Fig. 6c, d).

Notably, neither RNF5 overexpression nor RNF5 KD altered the abundance of RBBP4 protein, suggesting that RBBP4 ubiquitination by RNF5 does not occur via the formation of proteasome-targeting K48 ubiquitin chains and does not alter RBBP4 stability (Fig. 6f, g and Supplementary Fig. 6e). Immunoprecipitation of RBBP4 and immunoblot using an antibody specific for the K63 chain topology revealed no notable differences in cells overexpressing any form of RNF5, suggesting

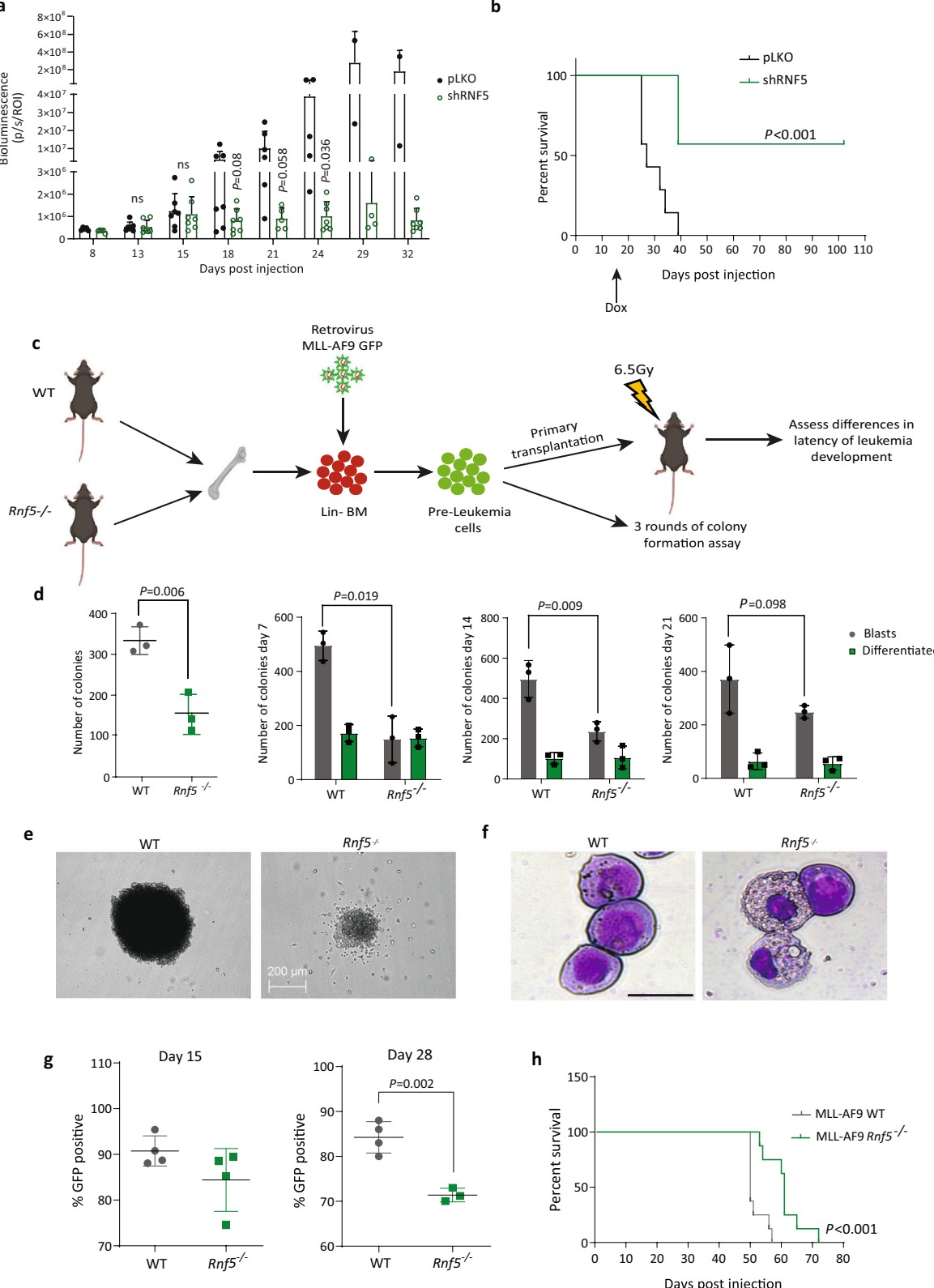

that RNF5 does not induce K63 ubiquitination of RBBP4 (Supplementary Fig. 6f). To assess the linkage-specific poly-ubiquitin induced by RNF5 on RBBP4, we used mutant HA-ubiquitin constructs with only one lysine available for linkage (K-only mutants: K29, K11, K27, K6, and K33) in which all lysine residues except that indicated are mutated to arginine allowing a single type homotypic chain. We then monitored changes in Myc-tagged RBBP4 ubiquitination in cells overexpressing Flag-tagged RNF5. Poly-ubiquitination of RBBP4 was enhanced by RNF5 only in the presence of K29 ubiquitin (Fig. 6h), strongly suggesting that RNF5 induces K29-topology polyubiquitination of RBBP4.

**Fig. 4 RNF5 suppression impairs leukemia establishment and progression in vivo. a** Graph depicting growth in mice of luciferase-expressing U937-pGFL cells transduced with empty vector (pLKO) or inducible shRNF5. Bioluminescence was quantified to monitor disease burden. Data are presented as the mean ± SD of 7 mice per group. *P* values were determined using unpaired two-tailed *t*-test. **b** Kaplan–Meier survival curves of mice injected with U937-pGFL cells expressing pLKO ($n = 7$ mice/group) or inducible shRNF5 ($n = 7$ mice/group). $P < 0.001$ by two-sided Mantel–Cox log-rank test. **c** Schematic representation of the experiment. Lin⁻Sca1⁺c-Kit⁺ (LSK) cells were purified from the bone marrow of WT or *Rnf5*⁻/⁻ mice, transduced in vitro with a GFP-tagged MLL-AF9 fusion gene, and then either analyzed by colony-forming assays in vitro or intravenously injected into sub-lethally irradiated WT C57BL/6 mice. **d** Quantification of total colonies (left) or blast-like and differentiated colonies (right) of GFP-MLL-AF9–transformed WT or *Rnf5*⁻/⁻ cells after 7, 14, or 21 days in culture. Data are presented as the mean ± SD of three independent experiments. *P* values were determined using a paired two-tailed *t*-test. **e** Representative pictures of colonies of GFP-MLL-AF9–transformed WT or *Rnf5*⁻/⁻ cells after 7 days in culture. Scale bar 200 μm. **f** Wright–Giemsa-staining of GFP-MLL-AF9–transformed WT or *Rnf5*⁻/⁻ cells after 7 days in culture. Scale bar 25 μm. **g** Flow cytometric quantification of GFP+ cells in peripheral blood of mice intravenously injected with GFP-MLL-AF9–transformed WT ($n = 4$) or *Rnf5*⁻/⁻ ($n = 4$) cells at days 15 and 28 post-injection. Data are presented as the mean ± SD. $P = 0.002$ by unpaired two-tailed *t*-test. The gating strategy is provided in Supplementary Fig. 8b. **h** Kaplan–Meier survival curves of mice injected with GFP-MLL-AF9–transformed WT or *Rnf5*⁻/⁻ cells. Data are from two independent experiments ($n = 4$ mice/group per experiment). $P < 0.001$ by two-sided Mantel–Cox log-rank test. Source data are provided as a Source Data file.

**RNF5 promotes recruitment of RBBP4 to gene promoters**. Because RNF5 activity does not alter RBBP4 stability, we asked whether RNF5 affects RBBP4 localization or interactions with other proteins. Subcellular fractionation in MOLM-13 cells and immunofluorescent analyses of nuclear and chromatin-bound RBBP4 did not identify changes in RBBP4 localization following RNF5 KD (Supplementary Fig. 6g and h). Next, since RBBP4 is a component of PRC2 and complexes containing HDACs[31,32], we asked whether RNF5 activity alters the formation of these complexes or their recruitment to target gene promoters. Neither overexpression nor KD of RNF5 affected RBBP4 interaction with HDAC1, HDAC2, or EZH2 (Fig. 6i, j and Supplementary Fig. 6i), suggesting that RBBP4 ubiquitination by RNF5 is not required for assembly of RBBP4-containing these complexes.

We then used chromatin immunoprecipitation (ChIP) and quantitative PCR (qPCR) to investigate RBBP4 recruitment to promoters of genes regulated by either RNF5 or RBBP4. RNF5 KD decreased RBBP4 recruitment to *ANXA1*, *NCF1*, and *CDKN1A* promotors (Fig. 6k). Examination of histone modifications at promoters of these genes identified that RNF5 KD increased H3K9 and H3K27 acetylation (Fig. 6l, m) and reduced H3K27 methylation (Fig. 6n), changes indicative of increased gene expression[33,34]. These changes are consistent with increased expression of these genes following RNF5 KD (Fig. 5c) and suggest that RNF5 control of gene expression in AML cells is mediated by RBBP4.

**RNF5 inhibition sensitizes AML cells to HDAC inhibitors**. As an independent support for the function of the RNF5-RBBP4 regulatory axis in promoting AML cell growth, we screened for synergistic interactions between RNF5 and epigenetic modulators. To do so, we assessed the effect of 261 epigenetic inhibitors at two concentrations (Supplementary Table 2) on the growth of U937 cells that stably express inducible shRNF5 (Fig. 7a, b). Of epigenetic inhibitors tested, 49 decreased viability of shRNF5-expressing cells relative to control WT RNF5 AML cells (Fig. 7b). Among the 49 inhibitors were several hypomethylation agents, including several histone methyltransferases (such as G9a), histone demethylases (such as Jumonji histone demethylases), and HDAC inhibitors (such as TMP269, pimelic diphenylamide 106, and *N*-acetyldinaline [CI-994]). Because RBBP4 is a key component of the HDAC complex and given that RNF5 KD induces transcriptional changes comparable to HDAC1 inhibition (Fig. 5d, e), we assessed possible synergy between RNF5 inhibition and HDAC inhibitors. To do so, we selected the HDAC inhibitor CI-994, which is in clinical trials against several cancers (https://www.drugbank.ca/drugs/DB12291), for additional validation. Indeed, U937 and HL-60 cells subjected to RNF5-KD were more sensitive to CI-994

relative to control cells (Fig. 7c and Supplementary Fig. 7a), suggesting that RNF5 KD sensitizes AML cells to HDAC inhibition.

We note that the HDAC inhibitor romidepsin (also known as FK228) did not score positively in our screen. This was likely due to the relatively high concentrations tested, which were lethal to both RNF5-KD and control U937 cells. FK228 has been approved by the Food and Drug Administration (FDA) to treat peripheral T-cell lymphoma[35] and has been investigated in preclinical studies as a potential treatment for AML[25,36]. Therefore, we re-assessed FK228 at non-lethal concentrations (up to 6 nM for 24 h) using multiple AML cell lines. Notably, when combined with RNF5-KD, FK228 had an additive effect in decreasing cell viability (Fig. 7d, e and Supplementary Fig. 7b–d) and inducing apoptosis (Fig. 7f). We also confirmed the additive effect of *RNF5* loss plus treatment with HDACi in MOLM-13 cells (made *RNF5* deficient using the CRISPR/Cas9 system; Supplementary Fig. 7e). The additive effect on AML cell death of RNF5-KD plus FK228 treatment was lost upon RNF5 re-expression (Fig. 7g), confirming a specific role for RNF5 in sensitizing AML cells to HDAC inhibition.

Of AML cell lines tested, the MV-4-11 line showed very low levels of RNF5 protein (Supplementary Fig. 1b). MV-4-11 cells were also most sensitive to FK228 treatment (Supplementary Fig. 7f), and RNF5 KD did not increase their sensitivity (Supplementary Fig. 7g). These observations further support the importance of RNF5 abundance in the response of AML cells to HDAC inhibitors. Moreover, RBBP4 KD sensitized AML cells to FK228 treatment (Fig. 7h and Supplementary Fig. 7h), consistent with our findings that RNF5 positively regulates RBBP4. Notably, elevated H3K9 acetylation at the promotors of RNF5- or RBBP4-regulated genes, such as *ANXA1* and *CDKN1A*, was seen following FK228 treatment and further increased upon RNF5 KD (Fig. 7i). The latter finding is consistent with increased *ANXA1* and *CDKN1A* expression seen after treatment with FK228 alone or in combination with RNF5 KD (Fig. 7j and Supplementary Fig. 7i). Notably, RNF5 KD in the K562 (CML) line did not sensitize cells to FK228, and RNF5 KD in Jurkat cells (T-ALL) only slightly enhanced cell sensitivity to FK228 (Supplementary Fig. 7j and k).

Next, to corroborate these findings in primary AML blasts, we performed an ex-*vivo* analysis of AML patients' samples ($n = 4$) and assessed their sensitivity to FK228 treatment. These samples were selected based on RNF5 and RBBP4 protein levels (2 high, 2 low). Interestingly, and similar to phenotypes seen in our KD experiments, samples expressing low RNF5 and RBBP4 were more sensitive (AML-075B logIC50 $= -10.7$ M, AML-034 logIC50 $= -10.4$ M) to FK228 treatment, compared with the high-expressing group (AML-013 logIC50 $= -9.9$ M, AML-072B logIC50 $= -9.6$ M) (Fig. 7k–m).

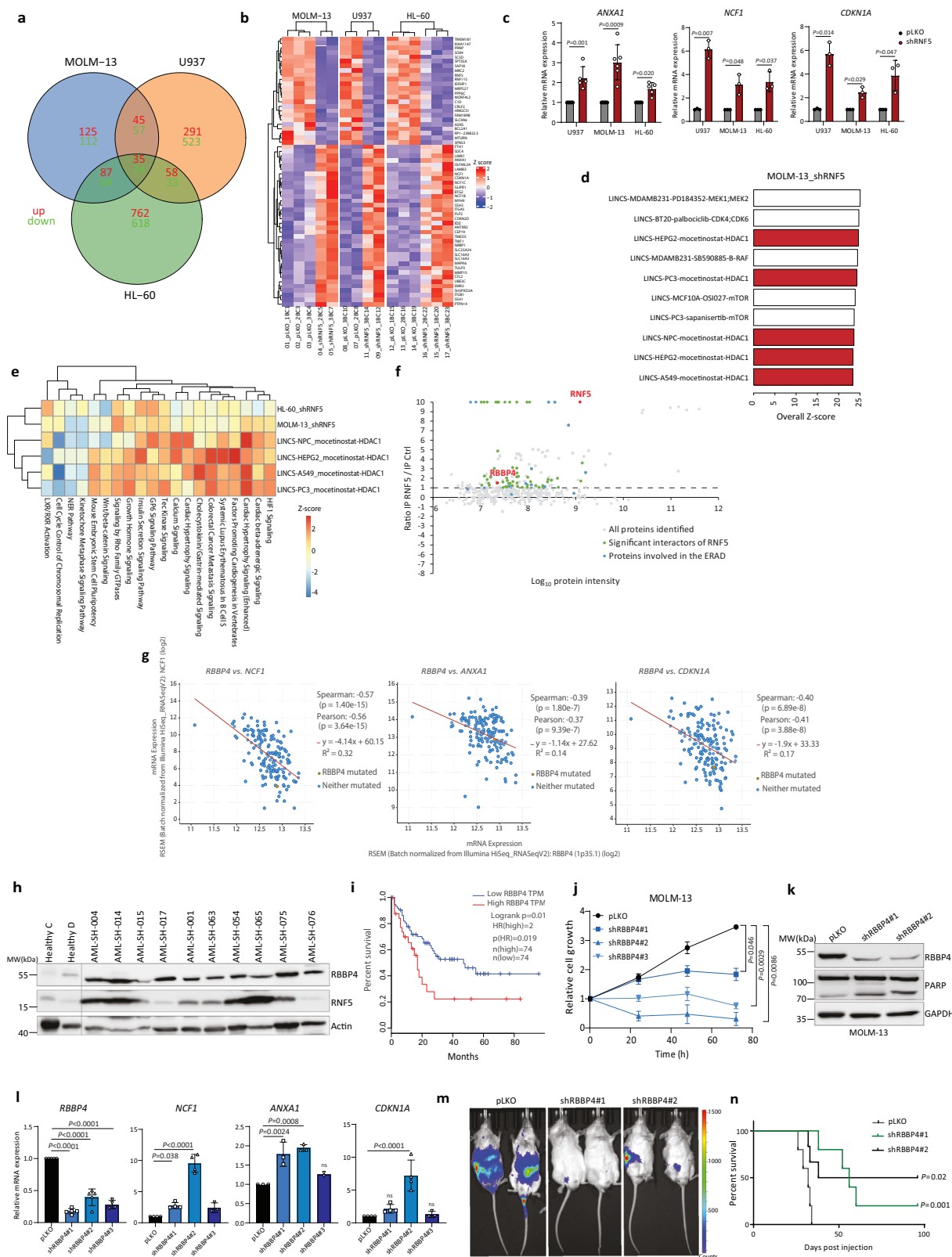

Finally, to assess the relevance of combined RNF5 loss and HDAC treatment in patients, we analyzed data from a bioinformatic pipeline that identifies clinically relevant synthetic lethal interactions[37]. This analysis revealed a more favorable prognosis in patients with concomitant downregulation of both *HDAC* and *RNF5* (Fig. 7n), substantiating the sensitization to HDAC in RNF5 low expressing specimens. Collectively, these findings suggest that

RNF5 signaling is a critical determinant of AML cell sensitivity to HDAC inhibitors.

## Discussion

High mortality seen in patients with AML predominantly results from failure to achieve complete remission following

**Fig. 5 Transcriptional and survival analysis of RNF5- or RBBP4-deficient AML cells. a** Venn diagram analysis of RNA-seq results showing upregulated (red) and downregulated (green) genes in AML lines following RNF5 KD. Overlapping areas indicate commonly modulated genes. **b** Heatmap of RNA-seq data performed on pLKO or shRNF5 AML lines, as indicated beneath maps (see Methods). **c** RT-qPCR validation of genes deregulated by RNF5-KD. Data are presented as the mean ± SD of n = 5 or 6 (*ANXA1*) or n = 3 (*NCF1* and *CDKN1A*) independent experiments. **d** Top ten drug screening results from the LINCS matched with transcriptomic data from the shRNF5 MOLM-13 line. Values are overall z-scores from the IPA Analysis Match database. HDAC1 inhibitor results are shown in red. **e** shRNF5 transduction of MOLM-13 or HL-60 promotes similar changes to those seen in NPC, HEPG2, A549, and PC3 cancer cell lines treated with the HDAC1 inhibitor mocetinostat. Z-score was calculated by IPA: A positive z-score predicts pathway activation; a negative z-score predicts inhibition. **f** Results of LC-MS/MS analysis present log$_2$-transformed ratio of proteins in anti-Flag immunoprecipitates of RNF5-overexpressing versus control cells. Green, proteins significantly enriched in RNF5-overexpressing cells; blue, enriched proteins in the ERAD pathway. RNF5 and RBBP4 are indicated in red. **g** Co-expression of *RBBP4* and indicated RNF5 target genes in AML samples analyzed in cBioPortal using the TCGA database (Pearson correlation, P < 0.0001, n = 165). **h** WB of RBBP4 in PBMCs from healthy subjects and AML patients (Scripps Health). **i** Overall survival rate (performed using GEPIA and TCGA) of AML patients expressing high (30%) or low (70%) levels of *RBBP4* transcripts. TPM: Transcripts Per Million. HR: hazard ratio. **j** Growth assay of MOLM-13 cells after transduction with indicated constructs. Data are presented as the mean ± SD of n = 3 independent experiments. **k** WB analysis of indicated proteins in MOLM-13 cells expressing indicated constructs. **l** RT-qPCR analysis of genes deregulated by RNF5-KD in MOLM-13 cells expressing the indicated constructs. Data are presented as the mean ± SD of n = 5 (*RBBP4*), n = 4 (*NCF1* and *CDKN1A*), or n = 3 (*ANXA1*) independent experiments. **m** Bioluminescent images of representative mice 4 weeks following transplantation of U937-pGFL expressing the indicated constructs. **n** Kaplan–Meier survival curves of mice injected with U937-pGFL cells expressing indicated vectors. P = 0.02 and P = 0.001 by two-sided Mantel–Cox log-rank test. P values were determined using paired two-tailed t-test (**c** and **l**) or two-way ANOVA followed by Tukey's multiple comparison test (**j**). Source data are provided as a Source Data file.

chemotherapy, coupled with a high relapse rate. Here we identify an important role for the ubiquitin ligase RNF5 in AML and demonstrate how RNF5 contributes to this form of leukemia. Our studies establish a function for RNF5 beyond its previously characterized activity in ERAD and proteostasis[6,30] and reveal mechanisms underlying its regulation of gene expression programs governing AML development and response to HDAC inhibitors. The clinical relevance of RNF5 and RBBP4 to AML is supported by our studies of patient samples and genetic mouse models. In mice, RNF5 or RBBP4 depletion inhibited AML progression and prolonged mouse survival (Fig. 4). In human, analysis of AML samples from two independent clinical cohorts revealed that a high abundance of RNF5 protein, which is commonly seen in AML patient samples, correlates with poor prognosis. Those phenotypes are mediated via RNF5 interaction with the chromatin remodeling protein RBBP4, which results in its non-canonical (K29 topology) ubiquitination that promotes RBBP4 recruitment to specific gene promoters (among them, *ANXA1*, *NCF1*, and *CDKN1A*), and a concomitant regulation of genes implicated in AML maintenance. Our finding that RNF5 modifies RBBP4 in a way that alters the expression of AML-related genes is confirmed by our ChIP analysis showing that RNF5 promotes recruitment of RBBP4 to gene promoters. Moreover, an inverse correlation between *RBBP4* expression and the expression of genes that were upregulated in RNF5 KD cells was also found in AML samples from the TCGA database (Fig. 5). Future genome-wide assessment of promoter-bound RBBP4 will likely identify additional genes whose transcription is regulated by RNF5-modified RBBP4.

Independent support for a function for RNF5 in recruiting RBBP4 to transcriptional regulatory complexes comes from our finding that RNF5/RBBP4 abundance governs the sensitivity of AML cells to HDAC inhibitors. Correspondingly, transcriptional changes induced by RNF5 KD overlapped with those seen following treatment with HDAC1 inhibitors. Furthermore, AML primary blasts expressing low RNF5/RBBP4 levels were more sensitive to FK228 compared to high-expressing blasts. Along these lines, synthetic lethal analysis identified a favorable prognosis in a cohort of AML patients with low expression of both *HDAC* and *RNF5* (Fig. 7). These findings suggest that RNF5 or RBBP4 abundance may serve as useful markers for the stratification of AML patients for treatment with HDAC inhibitors.

Notably, although RNF5 is expressed at high levels in AML, CML, and T-ALL cell lines[20] it is critical for cell survival only in

AML cells. In fact, the CCLE database reveals that CML and T-ALL lines express on average higher levels of *RNF5* than do AML lines. Nonetheless, K-562 (CML) and Jurkat (T-ALL) lines subjected to RNF5 KD do not exhibit growth inhibition or undergo cell death, while similarly treated AML lines do. Likewise, inhibition of RBBP4 does not impact CML or T-ALL cell but rather inhibits AML cell growth in a manner similar to that seen after RNF5 inhibition. Along these lines, RNF5 regulation and function are likely to be cell type and tissue-dependent. Our earlier studies in breast cancer and melanoma further support such context-dependent functions: RNF5 promotes melanoma growth via changes in immune and intestinal epithelial cells, while inhibits breast cancer growth through the tumor-intrinsic expression of glutamine carrier proteins[7,8,38].

It is important to note that relatively high RNF5 expression in AML cell lines is likely due to a high copy number, as shown by analysis of copy number alterations in various cancer cells[20]. Analysis of the TCGA database reveals increases in RNF5 mRNA levels in 3% of patients. Our patient cohorts revealed a significant increase in RNF5 abundance but not transcription levels (Fig. 1b, c and Supplementary Fig. 1e). We therefore believe that RNF5 overexpression (at the protein level) is attributable to relative increases in RNF5 protein stability, as supported by our assessments of two independent AML cohorts (Fig. 1). As a ubiquitin ligase, RNF5 activity is regulated primarily by post translational modifications affecting its stability, rather than transcription. However, that these increases may be linked to a pre-existing mutation that could increase RNF5 abundance in AML patient PMBCs or to micro-vesicle-based cell-cell communication cannot be ruled out.

Given that RNF5 protein is ER-anchored, its interaction with a chromatin regulatory protein such as RBBP4 is unanticipated. Among that possibilities that may explain such interaction are: (i) The interaction may occur as the *RBBP4* gene is translated, prior to nuclear translocation, a mechanism reported for other ubiquitin ligases[39], (ii) An as yet unidentified post-translational modification may promote nuclear localization of RNF5. For example, the possibility that RNF5 undergoes sumoylation should be considered given the high probability predicted using GPS-SUMO tool[40]; Finally, (iii) RBBP4/RNF5 interaction may occur at specific phases of the cell cycle, for example, at the entry to mitosis, when the nuclear envelop breaks down and nuclear contents are released to the cytoplasm. Future studies are needed to further examine these possibilities.

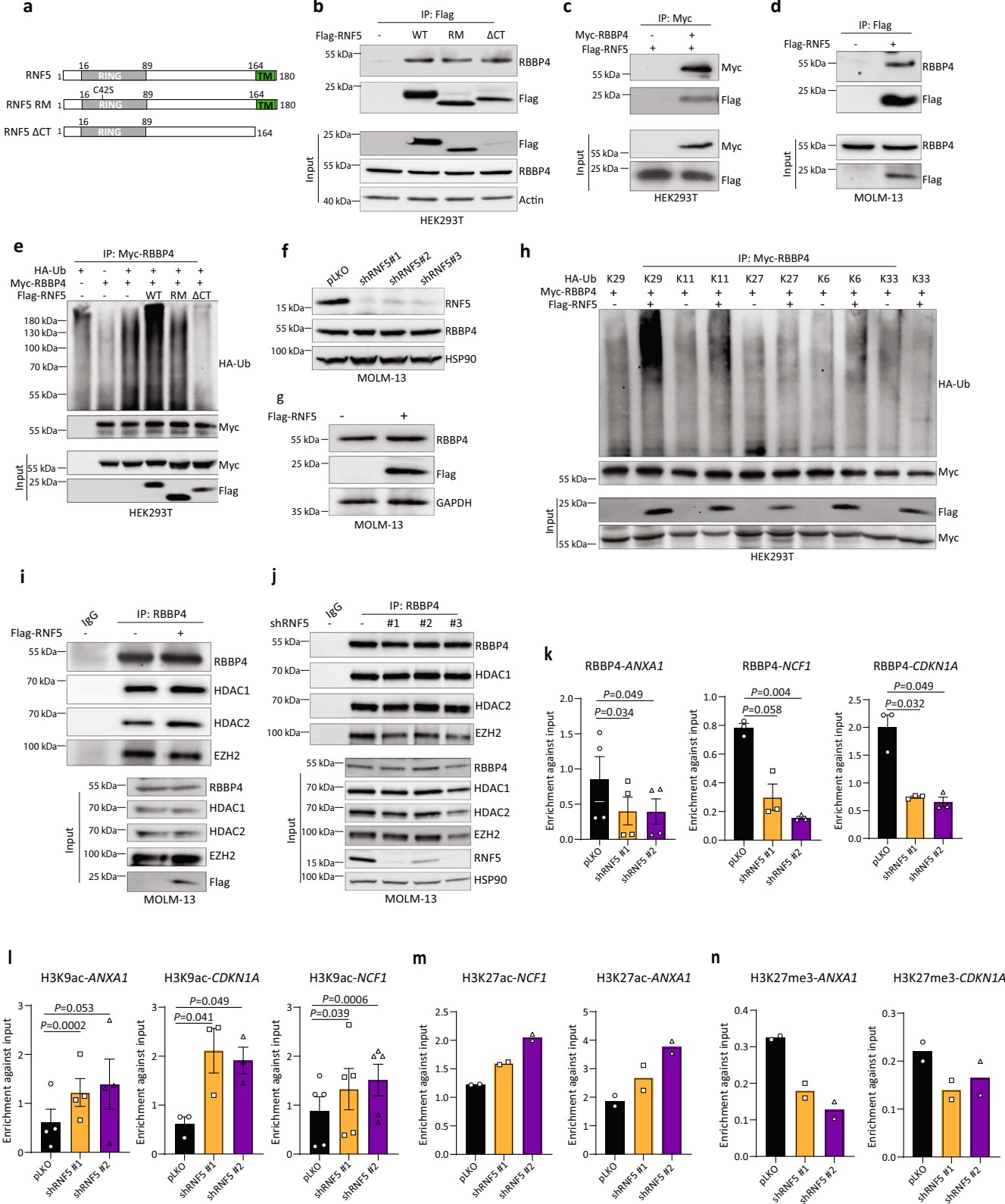

In summary, using genetic mouse models and clinical data, our findings establish a central role for the RNF5-RBBP4 axis in AML maintenance and responsiveness to HDAC inhibitors. Our identification of crosstalk between ubiquitination and epigenetic regulation offers a new paradigm for ERAD-independent RNF5 function in controlling RBBP4 activity and subsequent transcriptional networks implicated in AML. Our findings also demonstrate the ability of HDAC inhibitors to treat AML,

**Fig. 6 RBBP4 is ubiquitinated by RNF5 and regulates RNF5 target genes. a** Schematic showing full-length and mutants forms of RNF5.
**b** Immunoprecipitation (IP) and Western blot (WB) analysis of HEK293T cells transfected with Flag-tagged forms of full-length RNF5 (WT), the
catalytically inactive RING domain mutant (RM), or the C-terminal transmembrane domain deletion mutant (ΔCT). Cells were treated with MG132 (10 μm,
4 h) before lysis. **c** IP and WB of HEK293T cells co-expressing Myc-RBBP4 and Flag-tagged RNF5-WT treated with MG132 (10 μm, 4 h) before lysis. **d** IP
and WB of ectopically expressed doxycycline-inducible Flag-tagged RNF5 and endogenous RBBP4 in MOLM-13 cells. Cells were incubated for 2 days with
or without doxycycline (1 μg/ml) and with MG132 (10 μm, 4 h) before lysis. **e** WB of anti-Myc IP from lysates of HEK293T cells co-expressing Myc-RBBP4,
hemagglutinin-tagged ubiquitin (HA-Ub), and indicated Flag-tagged RNF5 constructs. Cells were treated with MG132 (10 μm, 4 h) before lysis. **f** WB of
indicated proteins in MOLM-13 cells expressing empty vector (pLKO) or indicated shRNF5 construct. **g** WB analysis of indicated proteins in MOLM-13 cells
expressing empty vector or doxycycline-inducible Flag-tagged RNF5. **h** WB of anti-Myc IP and lysates of HEK293T cells co-expressing Myc-RBBP4, Flag-
tagged RNF5, and different HA-tagged ubiquitin mutants (K29, K11, K6, K27, and K33). MG132 (10 μm, 4 h) was added before lysis. **i** IP and WB for the
interaction of RBBP4 with HDAC1, HDAC2, or EZH2 in MOLM-13 cells expressing indicated constructs. **j** IP and WB for RBBP4 interaction with HDAC1,
HDAC2, or EZH2 in MOLM-13 cells expressing indicated constructs. MG132 (10 μm, 4 h) was added before lysis. **k** ChIP and qPCR reveal the enrichment of
RBBP4 (normalized to input) at indicated gene promoters in MOLM-13 cells expressing indicated constructs. **l–n** ChIP and qPCR reveal the enrichment of
H3K9ac, H3K27ac, or H3K27me3 (normalized to input) at indicated gene promoters in MOLM-13 cells expressing indicated constructs. Data in **k** and **l** are
presented as mean ± SEM of $n = 4$ (*ANXA1*) or $n = 3$ (*NCF1* and *CDKN1A*) independent experiment. Data in **m** and **n** are the mean of $n = 2$ independent
experiments. The *P* values were determined using paired two-tailed *t*-test (**k** and **l**). Source data are provided as a Source Data file.

---

particularly AML expressing low levels of RNF5, and provides a method to stratify AML patients for treatment with HDAC inhibitors.

## Methods

**Animal studies**. All animal experiments were approved by the Sanford Burnham Prebys Medical Discovery Institute's Institutional Animal Care and Use Committee (approval AUF 16-028). Animal care followed institutional guidelines. *Rnf5*$^{-/-}$ mice were generated on a C57BL/6 background as described[23]. C57BL/6 WT mice were obtained by crossing *Rnf5*$^{-/-}$ mice. Female mice were maintained under controlled temperature (22.5 °C) and illumination (12 h dark/light cycle) conditions and were used in experiments at 6–10 weeks of age.

The xenograft model was established using U937 cells expressing the p-GreenFire1 Lenti-Reporter Vector (pGFL). NOD/SCID (NOD.CB17-Prkdcscid/J) mice were obtained from the SBP Animal Facility. Mice were irradiated (2.5 Gy), and U937-pGFL cells ($2 \times 10^4$ per mouse) were injected intravenously. Leukemia burden was serially assessed using noninvasive bioluminescence imaging by injecting mice intraperitoneally (i.p.) with 150 mg/kg D-Luciferin (PerkinElmer, 122799) in phosphate-buffered saline (PBS, pH 7.4), anesthetizing them with 2–3% isoflurane, and imaging them on an IVIS Spectrum (PerkinElmer). For in vivo RNF5 KD experiments, at disease onset (day 15, as measured by bioluminescent imaging), mice were fed rodent chow containing 200 mg/kg doxycycline (Dox diet, Bio-Serv) to induce RNF5-KD. Mice were sacrificed upon signs of morbidity resulting from leukemic engraftment (>10% weight loss, lethargy, and ruffled fur).

**Cell culture**. Human HEK293T and A375 cells were obtained from the American Type Culture Collection (ATCC). U937 and K562 cells were kindly provided by Prof. Yuval Shaked; Kasumi-1 cells were from Prof. Tsila Zuckerman; and MV-4-11, GRANTA, THP-1, and MEC-1 cells were from Dr. Netanel Horowitz. MOLM-13 cells were kindly provided by Dr. Ani Deshpande (SBP Discovery Institute, USA), KG-1a, HL-60, Jurkat, RPMI 8226, and HAP-1 cells were a kind gift from Prof. Ciechanover (Technion, Israel). MOLM-13, U937, THP-1, Kasumi, Jurkat, and RPMI-8226 cells were cultured in RPMI medium; HL-60, MV-4-11, K-562, MEC-1, HAP-1, and KG-1α cells were cultured in IMDM; and GRANTA, A375, and HEK293T cells were cultured in DMEM. All media were supplemented with 10% fetal bovine serum (FBS), 1% L-glutamine, penicillin (83 U/mL), and strep-tomycin (83 μg/mL) (Gibco). Cells were regularly checked for mycoplasma contamination using a luminescence-based kit (Lonza).

**Primary AML cells**. AML patient samples were obtained from Scripps MD Anderson, La Jolla, CA (IRB-approved protocol 13-6180), and written informed consent was obtained from each participant. Samples were also obtained from the Rambam Health Campus Center, Haifa, Israel (IRB-approved protocol 0372-17). Fresh blood samples were obtained by a peripheral blood draw, PICC line, or central catheter. Filgrastim-mobilized peripheral blood cells were collected from healthy donors and cryopreserved in DMSO. PBMCs were isolated by cen-trifugation through Ficoll-Paque™ PLUS (17-1440-02, GE Healthcare). Residual red blood cells were removed using RBC Lysis Buffer for humans (Alfa Aesar, cat. # J62990) according to the manufacturer's instructions. The final PBMC pellets were resuspended in Bambanker serum-free freezing medium (Wako Pure Chemical Industries, Ltd.) and stored under liquid N₂. Patients' information is provided in Supplementary Table 1.

MLL-AF9 patient-derived xenograft (PDX) samples (from the Jeremias Lab, Munich, Germany) were cultured in IMDM medium with 20% BIT (Stemcell Technologies), human cytokines, and StemRegenin 1 (SR1) and UM171, as described[41]. Cells were transduced with empty vector or different shRNF5

constructs as described below (see transfections and transduction section) and plated in 100 μL per well of a complete medium in 96-well plates. Growth was monitored every 24 h using the CellTiter Glo reagent.

For the drug dose responses, FK228 was diluted in DMSO at 10 mM and serially diluted (1/3, x13 concentrations) in a Labcyte Echo Low Dead Volume (LDV) plate. 25nLs of drugs at 1000x concentration were spotted in quadruplicate in 384-well plates (Greiner #781098) using a Labcyte Echo 550 acoustic dispenser, and patient AML cells (described above) were seeded (2.5 k cells/well in 25 μLs) onto 3 plates with a Multidrop Combi Reagent Dispenser (Thermo). After 2 days, cell viability was assessed by adding 10 μLs/well of CellTiter-Glo reagent (Promega #G7572) using a Multidrop Combi, and luminescence was read on an Envision plate reader (Perkin Elmer). Raw data were processed in Microsoft Excel, with cell viability values normalized to percentages relative to vehicle (0.1% DMSO) controls. Data were graphed and subjected to statistical analyses using GraphPad Prism software (v.9.1.1).

**Antibodies and reagents**. The RNF5 antibody was produced as described previously (1:1000)[7,23]. Other antibodies used were: rabbit anti-cleaved caspase-3 (#9661, Cell Signaling Technology, 1:1000), rabbit anti-PARP (#9532, Cell Signaling Technology, 1:1000), mouse anti-RBBP4 (NBP1-41201, Novus Biologicals, 1:500), mouse anti-glyceraldehyde 3-phosphate dehydrogenase (GAPDH; ab8245, Abcam, 1:10000), mouse anti-Tubulin (T9026, Sigma, 1:5000), mouse anti-Flag (F1804, Sigma, 1:2000), mouse anti-Myc-Tag (#2276, Cell Signaling Technology, 1:1000), mouse anti-HA (901501, Biolegend, 1:2000), rabbit anti-HDAC1 (#2062, Cell Signaling Technology, 1:1000), rabbit anti-HDAC2 (57156, Cell Signaling Technology, 1:1000), rabbit anti-Ezh2 (5246, Cell Signaling Technology, 1:1000), mouse anti-HSP90 (sc-13119, Santa Cruz Bio-technology, 1:1000), rabbit anti-p27, (#3688, Cell Signaling Technology, 1:1000), rabbit anti-p21 (#2947, Cell Signaling Technology, 1:1000), mouse anti-Ubiquitin (#3939, Cell Signaling Technology, 1:1000), rabbit anti-K63-linkage Specific Polyubiquitin (#5621, Cell Signaling Technology, 1:1000), rabbit anti-Actin (#4970, Cell Signaling Technology, 1:1000), rabbit anti-Histone H3 (#9717, Cell Signaling Technology, 1:1000), mouse anti-Caspase-3 (sc-56053, Santa Cruz Biotechnology, 1:1000), and mouse anti- Calregulin (sc-166837, Santa Cruz Biotechnology, 1:1000). HRP-conjugated secondary antibodies were from Jackson ImmunoResearch (goat-anti-mouse-HRP (AB_2338504) and goat-anti-rabbit-HRP (AB_2337938) and diluted 1:10000.

Romidepsin and *N*-acetyldinaline were purchased from Cayman Chemicals. Thapsigargin and tunicamycin were purchased from Sigma-Aldrich. MG132 was obtained from Selleckchem. Puromycin was purchased from Merck. Annexin V-APC and propidium iodide were from BioLegend.

**Plasmids and constructs**. Plasmids expressing Flag-RNF5-WT, Flag-RNF5-RM, and Flag-RNF5-ΔCT were described previously[5,7]. To generate doxycycline-inducible RNF5-WT, RNF5-RM, and RNF5-ΔCT overexpression vectors, coding sequences were amplified by PCR from pCDNA3.1-RNF5-WT, pCDNA3.1-RNF5-RM, and pCDNA3.1-RNF5-ΔCT, respectively, and the product was inserted into EcoRI-linearized pLVX TetOne-puro plasmid (Clontech) using the NEBuilder HiFi Assembly kit (New England BioLabs). Expression vectors encoding Myc-RBBP4 (#20715), HA-Ubiquitin (#18712), and HA-ubiquitin mutants (including K6 (#22900), K11 (#22901), K27 (#22902), K29 (#22903), and K33 (#17607)) were obtained from Addgene.

**Gene silencing**. Lentiviral pLKO.1 vectors expressing RNF5 or RBBP4-specific shRNAs were obtained from the La Jolla Institute for Immunology RNAi Center (La Jolla, CA, USA). Sequences were: shRNF5 #1 (TRCN0000004785) GAGTGTCCAG-TATGTAAAGCT, shRNF5 #2 (TRCN0000004788) CGGCAAGAGTGTCCAGT ATGT, shRNF5 #3 GAGGATGGATTGAGAGAAT, and inducible shRNF5, which has the same sequence as shRNF5#1. Sequences for RBBP4-specific shRNAs were: shRBBP4#1 (TRCN0000286103) GCCTTTCTTTCAATCCTTATA, shRBBP4#2

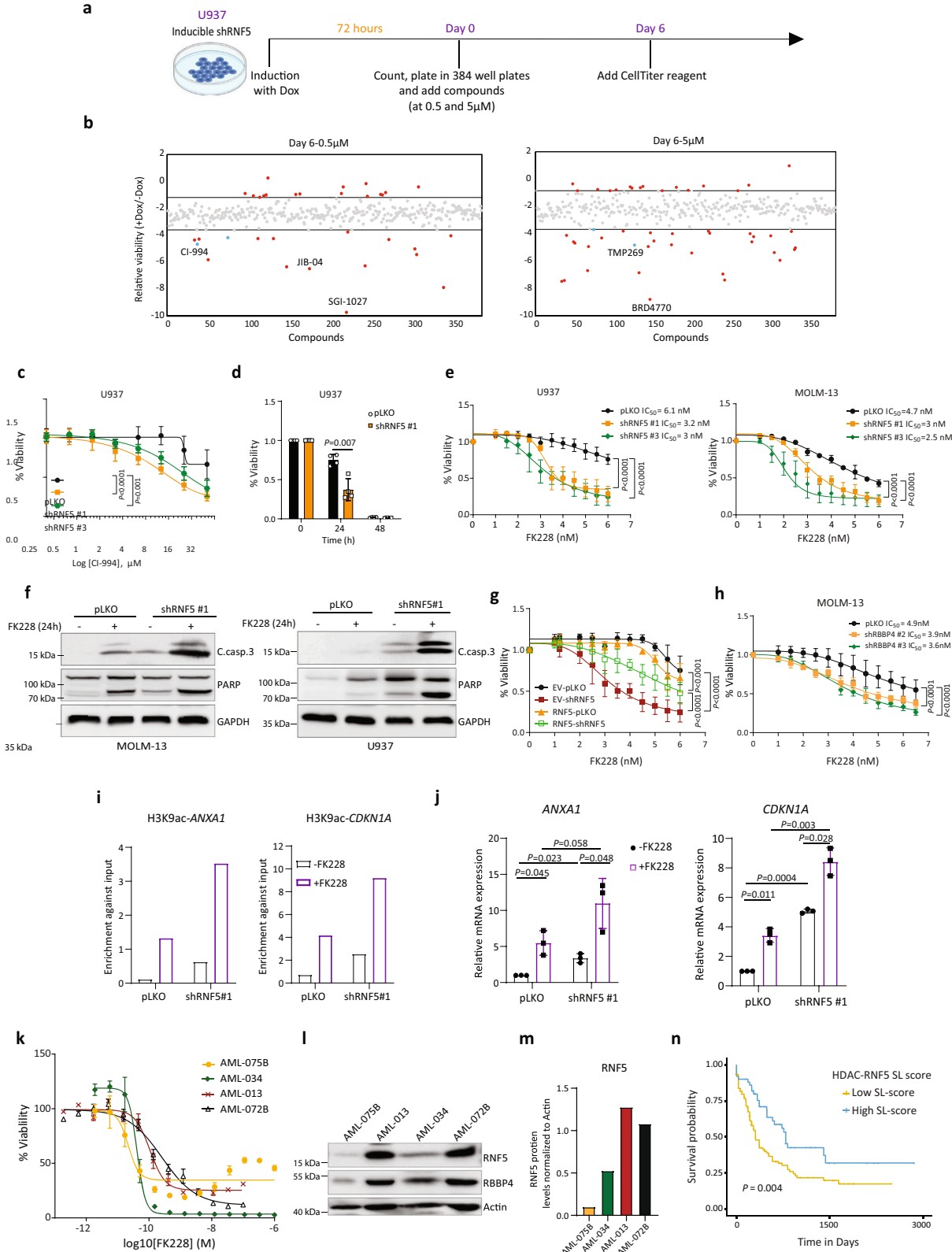

(TRCN0000293556) TGGTCATACTGCCAAGATATC, shRBBP4#3
(TRCN0000293554) ATGCGTCACACTACGACAGTG.

**Transfections and transduction.** Transient transfections were carried out using CalFectin (SignaGen) according to the manufacturer's recommendations.

Lentiviral particles were prepared using standard protocols. In brief, HEK293T cells were transfected with relevant vectors and the second-generation packaging plasmids ΔR8.2 and Vsv-G (Addgene). Virus-containing supernatants were collected 48 h later and then added in the presence of Polybrene to AML cells pre-seeded at ~5 × 10^5/well in 24-well plates (Sigma-Aldrich). After 8 h, cells were transferred to 10 cm culture dishes for an additional 24 h prior to experiments.

**Fig. 7 RNF5 inhibition sensitizes AML cells to HDAC inhibitors. a** Schematic showing the experimental design of the epigenetic screen. **b** Log₂-transformed ratios of the relative viability of doxycycline-induced (+Dox) versus uninduced (-Dox) U937 cells treated with compounds for 6 days. Red dots represent compounds that altered viability of RNF5-KD more than of uninduced cells, blue dots represent candidate HDAC inhibitors, and grey dots represent the remaining compounds tested. **c** Viability of U937 cells expressing indicated constructs after treatment for 24 h with CI-994. **d** U937 cell viability after treatment with 3.5 nM FK228. **e** Viability of U937 cells or MOLM-13 cells expressing indicated constructs 24 h following FK228 treatment. **f** WB of apoptotic markers in MOLM-13 and U937 cells expressing indicated constructs and incubated with or without FK228 (4 nM for 24 h). **g** Viability of U937 cells expressing indicated constructs and treated for 24 h with FK228. EV-pLKO, control cells; EV-shRNF5, cells expressing empty vector and shRNF5; RNF5-pLKO, cells overexpressing RNF5 and pLKO vector; RNF5-shRNF5, cells overexpressing RNF5 and shRNF5. **h** Viability of MOLM-13 cells expressing indicated constructs 24 h following FK228 treatment. **i** ChIP and qPCR indicating H3K9ac enrichment (normalized to input) at indicated gene promoters in MOLM-13 cells expressing indicated constructs. Data are presented as the mean ± SD of two independent experiments. **j** RT-qPCR analysis of indicated genes in MOLM-13 cells expressing indicated constructs following FK228 treatment (4 nM, 15 h). **k** Viability assay of 4 primary AML blasts (Scripps Health) 48 h following FK228 treatment. **l** WB analysis of RNF5 and RBBP4 in AML patient samples used for the ex-*vivo* drug analysis in (**l**). **m** WB quantification of RNF5 protein levels in (**l**) normalized to actin. **n** Kaplan–Meier plot showing survival analysis of AML patients segregated based on a median synthetic lethality (SL) score. Co-occurrence of low HDAC and RNF5 transcript levels in a patient's tumor (high SL score; blue line), compared with the rest of the patients (low score, yellow line). Data are presented as the mean ± SD of n = 4 (**c** and **d**), n = 3 (**e**, **g**, and **j**), or n = 5 (**h**) experiments. *P* values were determined using paired two-tailed *t*-test (**d** and **j**) or two-way ANOVA followed by Tukey's multiple comparison test (**c**, **e**, **g**, and **h**). Source data are provided as a Source Data file.

**Western blotting**. Cells were washed twice with cold PBS and lysed by addition of Tris-buffered saline (TBS)-lysis buffer (TBS [50 mM Tris-HCl pH 7.5, 150 mM NaCl], 0.5% Nonidet P-40, 1× protease inhibitor cocktail [Merck], and 1× phosphatase inhibitor cocktail[42] followed by incubation on ice for 20 min. Blood cells from healthy control subjects and AML patients were lysed using hot lysis buffer [100 mM Tris-HCl pH 7.5, 5% sodium dodecyl sulfate (SDS)] followed by incubation 5 min at 95 °C and sonication. Some samples were subjected to fractionation using a subcellular protein fractionation kit (Thermo Scientific Pierce), as indicated. Samples were resolved on SDS-PAGE and transferred to nitrocellulose membranes. Membranes were incubated for 1 h at room temperature with blocking solution (0.1% Tween 20/5% non-fat milk in TBS) and then overnight at 4 °C with primary antibodies. Membranes were washed with TBS and incubated for 1 h at room temperature with appropriate secondary antibodies (Jackson ImmunoResearch). Finally, proteins were visualized using a chemiluminescence method (Image-Quant LAS400, GE Healthcare, or ChemiDoc MP imaging system, Bio-Rad). The uncropped scans for all western blot are provided in the Source Data file.

**Immunoprecipitation**. Cells were lysed in TBS-lysis buffer as described above, centrifuged for 10 min at 17,000 *g*, and incubated overnight at 4 °C with appropriate antibodies. Protein A/G agarose beads (Santa Cruz Biotechnology) were then added for 2 h at 4 °C. Beads were pelleted by centrifugation, washed five times with TBS-lysis buffer, and boiled in Laemmli buffer to elute proteins. Finally, proteins were resolved on SDS-PAGE and subjected to Western blotting as described above.

**LC-MS/MS**. MOLM-13 cells were infected with doxycycline-inducible Flag-tagged RNF5-encoding or empty plasmids and expression were induced by the addition of doxycycline (1 μg/mL) for 48 h. The proteasome inhibitor MG132 (10 μM) was added for 4 h prior to harvest. Cells were lysed in TBS-lysis buffer, and total cell lysates were incubated with anti-Flag-M2-agarose beads (Sigma-Aldrich) overnight at 4 °C. Beads were washed with TBS-lysis buffer, and proteins were eluted from beads by addition of 3×Flag peptides (150 μg/mL, Sigma) for 1 h at 4 °C and then subjected to tryptic digestion followed by LC-MS/MS, as described[43].

Raw data were analyzed using MaxQuant (v1.5.5.0)[44] with default settings. Protein intensities were normalized using the median centering method. Fold-changes were calculated by dividing the protein intensity of Flag immunoprecipitates from RNF5-overexpressing cells by the protein intensity of Flag immunoprecipitates from control cells. Thresholds were set at 2 for fold-change and 0.05 for *p* value obtained using a two-sided Welch's *t*-test. Proteins identified in all RNF5 immunoprecipitation replicates but in one or no control IP replicates were considered potential RNF5 interactors if their corresponding fold-change was at least 2. Data from the Crapome (version 2.0)[42] repository were downloaded to filter potential contaminants. Cytoscape (version 3.8.1)[45] was used to generate the RNF5 interaction network and pathway enrichment analysis. Raw MS data were deposited in the MassIVE repository under the accession code MSV000083160.

**Immunofluorescence microscopy**. Cells were placed on coverslips on glass slides using a StatSpin cytofuge and fixed with 4% paraformaldehyde for 20 min at room temperature. Slides were then rinsed three times in PBS, and cells were permeabilized in 0.2% Triton X-100 in PBS for 5 min and blocked with 0.2% TX-100/10% FBS in PBS for 30 min. Primary antibodies were diluted in staining buffer (0.2% Triton X-100/2% FBS in PBS) and added to cells, and the slides were then incubated overnight at 4 °C in a humidified chamber. Slides were then washed three times in staining buffer, and secondary antibodies (Life Technologies) were diluted in staining buffer and added to slides for 1 h at room temperature in a humidified chamber shielded from light. Finally, slides were washed three times in staining buffer and mounted with Fluoroshield Mounting Medium containing 4′, 6-diamidino-2-phenylindole (DAPI; Sigma-Aldrich). Cells were analyzed using a fluorescence microscope (DMi8; Leica) with a 60× oil immersion objective. Images were processed using the 3D deconvolution tool from LASX software (Leica), and the same parameters were used to analyze all images.

**Cell viability assay**. Cell viability and growth were assayed using the CellTiter Glo kit (Promega) according to the manufacturer's recommendations. Cell lines were plated in white 96-well clear-bottomed plates (Corning) at a density of 7 × 10³ cells/well, and growth was monitored every 24 h using CellTiter Glo reagent. Viability was quantified by measuring luminescence intensity with an Infinite 2000 Pro reader (Tecan).

**Cell cycle analysis**. The distribution of cells in each phase of the cell cycle was analyzed by propidium iodide staining (Merck). Briefly, 1 × 10⁶ cells were washed twice with cold PBS and fixed in 70% ethanol in PBS at 4 °C overnight. Cells were washed, pelleted by centrifugation, and treated with RNase A (100 μg/mL) and propidium iodide (40 μg/mL) at room temperature for 30 min. Cell cycle distribution was assessed by flow cytometry (BD LSRFortessa™, BD Biosciences), and data were analyzed using FlowJo software.

**Annexin V and propidium iodide staining**. Cells were collected in FACS tubes, washed twice with ice-cold PBS, and resuspended in 100 μL PBS. Annexin V-APC (1.4 μg/mL) was added for 15 min at room temperature in the dark. Then, cells were washed in PBS and resuspended in 200 μL PBS, and then propidium iodide (50 μg/mL) was added. Finally, samples were analyzed by flow cytometry (BD LSRFortessa™, BD Biosciences). The gating strategy is provided in Supplementary Fig. 8c.

**Colony-forming assays**. For the soft agar assay, a base layer was formed by mixing 1.5% soft agar (low-melting-point agarose, Bio-Rad) and culture medium at a 1:1 ratio and placing the mixture in 6-well plates. Cells were resuspended in a medium containing 0.3% soft agar and added to the base layer at 1 × 10⁴ (MOLM-13) or 5 × 10³ (U937) cells/well. Agar was solidified by incubation at 4 °C for 10 min before incubation at 37 °C. Plates were incubated at 37 °C in a humidified atmosphere for 12–18 days. Cells were then fixed overnight with 4% paraformaldehyde, washed with PBS, and stained with 0.05% crystal violet (Merck) for 20 min at room temperature, and washed again with PBS. Plates were photographed and colonies were counted on the captured images.

For the methylcellulose assay, WT or *Rnf5*⁻/⁻ Lin⁻Sca1⁺c-Kit⁺ cells transformed with GFP-MLL-AF9 were resuspended in methylcellulose M3234 (Stem Cell Technologies) supplemented with 6 ng/mL IL-3, 10 ng/mL IL-6, and 20 ng/mL stem cell factor (PeproTech). Cells were then added to 35 mm dishes at 10³ cells/well and incubated for 6–7 days. Colonies were classified as compact and hypercellular (blast-like) or small and diffuse (differentiated). Virtually all colonies fell into one of these two categories.

**RT-qPCR analysis**. RNA was extracted using a GenElute Mammalian Total RNA Purification Kit (Sigma) according to standard protocols. RNA concentration was measured using a NanoDrop spectrophotometer (ThermoFisher). cDNA was synthesized from aliquots of 1 μg total RNA using a qScript cDNA Synthesis Kit (Quanta). Quantitative PCR was performed with SYBR Green I dye master mix (Roche) and a CFX connect Real-Time PCR System (Bio-Rad). Primer sequences are listed in Supplementary Table 3. Primer efficiency was measured in preliminary experiments, and amplification specificity was confirmed by dissociation curve analysis.

**Gene targeting using CRISPR/Cas9**. RNF5 sgRNAs were cloned into the pKLV2-U6gRNA-(BbsI)-PGKpuro2ABFP-W lentiviral expression vector and transduced into Cas9-expressing cell lines. All gRNAs were cloned into the BbsI site of the gRNA expression vector as previously described[46]. Briefly, HEK293T cells were co-transfected with pKLV2-U6gRNA-(BbsI)-PGKpuro2ABFP-W and ectopic packaging plasmids using CalFectin transfection reagent (SignaGen). Virus-containing supernatants were collected 48 h later. MOLM-13 cells were infected by the addition of supernatants for 48 h. Cells were then selected with puromycin (0.5 μg/mL) for 48 h and viability was measured. The RNF5-targeting sgRNA sequences were: sgRNF5 #3 F-GCACCTGTACCCCGGCGGAA and R-TTCCGCCGGGGT ACAGGTGC, and sgRNF5 #4 F-GTTCCGCCGGGGTACAGGTG and R-CAC CTGTACCCCGGCGGAAC.

**RNA-seq analysis**. PolyA RNA was isolated using the NEBNext Poly(A) mRNA Magnetic Isolation Module, and bar-coded libraries were constructed using the NEBNext Ultra™ Directional RNA Library Prep Kit for Illumina (NEB, Ipswich, MA). Libraries were pooled and single end-sequenced ($1 \times 75$) on the Illumina NextSeq 500 using the High output V2 kit (Illumina, San Diego, CA). Quality control was performed using Fastqc (v0.11.5, Andrews S. 2010), reads were trimmed for adapters, low quality 5′ bases, and the minimum length of 20 using CUTADAPT (v1.1). The number of reads per sample and the number of aligned reads suggested that read quality and number were good and that the data were valid for analysis. High-quality data were then mapped to a human reference genome (hg19) using the STAR mapping algorithm (version 2.5.2a)[47]. Feature Counts implemented in Subread (v1.50)[48] were used to count the sequencing reads from mapped BAM files. Analyses of differentially expressed genes were subsequently performed using a negative binomial test method (edgeR, v3.34.0)[49] implemented using SARTools R Package (v1.2.0)[50]. A list of the differentially expressed genes was exported into an excel file (Supplementary Data file 1), and pathway analysis was performed by uploading the lists of differentially expressed genes to IPA (http://www.ingenuity.com) using the following criteria: |log2 (fold-change)| > 0.4 and P value < 0.05. P values were determined using "Negative Binomial Generalized Linear Model (two-sided)" to generate the DEGs list. Multiple comparisons were also applied based on the "Benjamini & Hochberg" method. LINCS database[51] and other public data sets were processed by IPA. Molecular signatures for canonical pathways, upstream regulators, and causal networks were generated for each data set by IPA. Enrichment results in this study were compared to the LINCS molecular signatures by Analysis Match using z-scores developed by IPA. The z-scores represent how well activated or inhibited entities match data sets (% similarity). The top matched experiments in LINCS were selected by ranking the overall z-scores.

**Chromatin immunoprecipitation (ChIP)**. ChIP analysis was performed using a ChIP Assay Kit (Millipore) following the manufacturer's instructions. Briefly, $1 \times 10^6$ cells were used for each reaction. Cells were fixed in 1% formaldehyde at 37 °C for10 min, and nuclei were isolated with nuclear lysis buffer (Millipore) supplemented with a protease inhibitor cocktail (Millipore). Chromatin DNA was sonicated and sheared to a length between 200 bp and 1000 bp. Sheared chromatin was immunoprecipitated at 4 °C overnight with anti-H3K9ac (9649, Cell Signaling Technology), anti-H3K27ac (ab3594, Abcam), anti-H3K27me3 (9733, Cell Signaling Technology), and anti-RBBP4 (NBP1-41201, Novus). IgG was used as a negative control and anti-RNA polII (Millipore) served as a positive control antibody. Protein A/G bead-antibody/chromatin complexes were washed with low salt buffer, high salt buffer, LiCl buffer, and then TE buffer to eliminate nonspecific binding. Protein/DNA complexes were reverse cross-linked, and DNA was purified using NucleoSpin®. Levels of ChIP-purified DNA were determined by qPCR (see Supplementary Table 4 for primer sequences). Relative enrichment of the indicated DNA regions were calculated using the Percent Input Method according to the manufacturer's instructions and are presented as % input.

**Small molecule epigenetic regulator screen**. Aliquots of compounds (10 mM in DMSO) from a library of 261 epigenetic regulators were dispensed at final concentrations of 0.5 μM or 5 μM into the wells of a Greiner (Monroe, NC, Cat #781080) 384-well TC-treated black plate using a Labcyte Echo 555 acoustic pipette (Labcyte, San Jose, CA). U937 cells expressing an inducible shRNF5 vector were induced with doxycycline for 72 h and dispensed into the prepared plates at a density of $5 \times 10^2$/well in 50 μL RPMI-based culture medium (described above) using a Multidrop Combi (Thermo Fisher Scientific, Pittsburgh, PA). Plates were briefly centrifuged at ~100 g and incubated at 37 °C with 5% $CO_2$ for 6 more days using MicroClime Environmental lids (Labcyte, San Jose, CA). Plates were placed at room temperature for 30 min to equilibrate, 20 μL/well CellTiter-Glo Luminescent Cell Viability Assay reagent (Promega, Madison, WI) was added using a Multidrop Combi, and plates were analyzed with an EnVision multimode plate reader (PerkinElmer, Waltham, MA).

For the analysis, the intensity of induced shRNF5-expressing cells was divided by the intensity of uninduced cells. Ratios were log2 transformed and thresholds were calculated based on the distribution of the log2 ratios. The upper threshold was calculated as the Q3 + 1.5xQ, where Q3 is the third quartile and IQ is interquartile. The lower threshold was calculated as the Q1 − 1.5xIQ, where Q1 is the first quartile. Ratios outside these thresholds were considered outliers from the global ratio distribution and thus were potential candidates for having a differential effect on RNF5-KD or control cells.

**MLL-AF9–mediated transformation of bone marrow cells and generation of MLL-AF9–leukemic mice**. HEK293T cells were co-transfected with Murine Stem Cell Virus (MSCV)-based MLL-AF9 IRES-GFP[22] and ectopic packaging plasmids. Viral supernatants were collected 48 h later and added to Lin−Sca-1+c-Kit+ cells isolated from the bone marrow of WT or $Rnf5^{-/-}$ C57BL/6 mice. Transduced cells were maintained in DMEM supplemented with 15% FBS, 6 ng/mL IL-3, 10 ng/mL IL-6, and 20 ng/mL stem cell factor, and transformed cells were selected by sorting for GFP+ cells. To generate "primary AML mice," GFP-MLL-AF9–transduced cells were resuspended in PBS at $1 \times 10^6$ cells/200 μL and injected intravenously into sublethally irradiated (650 Rad) 6- to 8-week-old C57BL/6 female mice.

**Statistical analysis**. Differences between two groups were assessed using two-tailed unpaired or paired t-tests or Wilcoxon rank-sum test, and differences between group means were evaluated using t-tests or ANOVA. Two-way ANOVA with Tukey's multiple comparison test was used to evaluate experiments involving multiple groups. Survival was analyzed by the Kaplan–Meier method and evaluated with a log-rank test. All data were analyzed using GraphPad Prism version 8 or 9 (GraphPad, La Jolla, CA, USA) and expressed as means ± SD or SEM. P < 0.05 was considered significant. NS stands for not statistically significant.

**Reporting summary**. Further information on research design is available in the Nature Research Reporting Summary linked to this article.

## Data availability
The RAW MS data have been deposited in the MassIVE repository under accession code MSV000083160 (https://massive.ucsd.edu/ProteoSAFe/dataset.jsp?task=321eefef71fe4baa8900da284d5f66f3). RNA-Seq RAW data in FASTQ format from RNF5 knockdown experiments have been deposited in the NCBI Gene Expression Omnibus (GEO) database under accession code GSE155929. The raw data of TCGA database are available through GDC Data Portal (https://portal.gdc.cancer.gov/). The raw data of LINCS are available through NIH LINCS (https://lincsproject.org/). All other data are available within the article and its Supplementary Information. Source data are provided with this paper.

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

## Acknowledgements

We thank Drs. Yuval Shaked, Tsila Zuckerman, and Netanel Horowitz (Faculty of Medicine, Technion) for providing leukemic cell lines, and members of the Deshpande and Ronai labs for technical support and discussions. We thank SBP and Technion Core facilities for help along with the different phases of this study. We thank Nancy R. Gough (BioSerendipity, LLC) for editorial assistance. Z.A.R. gratefully acknowledges support from the National Cancer Institute grant (R35CA197465) and the Technion. A.K. was supported by a Faculty of Medicine fellowship at the Technion. Sanford Burnham Prebys Shared Resources are supported by an NCI Cancer Center Support Grant (P30 CA030199).

## Author contributions

A.K. and Z.A.R. designed the experiments; A.K., A.D., Yo.F., D.F, I.L., Y.L., and Id.L. performed experiments; A.K., J.S.L., B.F., D.F., Y.F., T.Z., J.Y., K.B., E.R., A.J.D, and Z.A.R. analyzed the data; I.P. and M.J. performed the epigenetic screen; D.F., I.J, R.A., N. H., Y.O., C.B., J.R.M., and K.V. provided access to patient samples and data; and A.K. and Z.A.R. wrote the paper.

## Competing interests

ZAR is co-founder and serves as scientific advisor to Pangea Therapeutics. All other authors declare no competing interests.
