## [Peer Review File · Nature Communications]

The ubiquitin ligase RNF5 Determines Acute Myeloid Leukemia Growth and Susceptibility to Histone Deacetylase InhibitorsEditorial Note: This manuscript has been previously reviewed at another journal that is not operating a transparent peer review scheme. This document only contains reviewer comments and rebuttal letters for versions considered at *Nature Communications*.

REVIEWERS' COMMENTS

Reviewer #2 (Remarks to the Author):

The authors have addressed most of the raised concerns.

Reviewer #3 (Remarks to the Author):

In the present revised manuscript, the authors found that increased expression of the ubiquitin ligase RNF5 contributes to AML development.

The manuscript might have prognostic and therapeutic applications.

Given that the over-expression of RNF5 in cancer was previously reported *Cancer Res* 2007 Sep 1;67(17):8172-9. doi: 10.1158/0008-5472.CAN-07-0045 this study originally identifies the axis RNF5/RBBP4 and its potential as prognostic role.

In this revised version the authors have improved the evidences in support of their hypothesis. Despite some points stay unanswered (as an example i) RNF5-RBBP4 in HDACi-resistant unresponsive cells;ii) global genomic assessment of RBBP4-dependent changes) the medical potential of the study is clear and so are the following applications.

Reviewer #4 (Remarks to the Author):

In this paper, the authors identify high protein levels of the E3 ligase RNF5 as being associated with a poor prognosis in AML. They go on to show that RNF5 knockdowns reduce leukaemia growth both in vitro and in vivo. Importantly, they show that wild type RNF5 expression can rescue the knockdowns while an enzymatic mutant of RNF5 fails to rescue the growth defect. Using an MLL-AF9 retroviral transduction model, they also show that transduced cells from *Rnf5*^{-/-} mice produce a leukaemia with a longer latency in an engraftment experiment. They also use the LINCS database to identify an overlapping gene expression signature between cells treated with HDAC inhibitors and those that harbour an RNF5 knockdown. A mass spec experiment allowed them to identify RNF5 interacting proteins, and they performed a thorough characterization of one factor, RBBP4. The main result from this section was that RNF5 mediated ubiquitination of RBBP4 had no impact on stability or localization in the cell, but did alter chromatin binding and loss of RNF5 protein levels caused increased histone acetylation at a defined subset of RBBP4 gene targets. Finally, in the last section of the paper, using an inhibitor screen they show that HDAC inhibitors synergize with RNF5 knockdowns, and they characterize one HDAC inhibitor in some detail. The overall conclusion is that low RNF5 protein levels could be a general predictor of HDAC inhibitor sensitivity.

I did not review the first version of this paper, but overall I think this is quite a comprehensive and interesting study. I read over the reviewer's comments, and in my opinion the authors have addressed many of the main points raised in previous reviews. I just have two minor points below.

Minor points

1) Line 334, "HDAC2, or EZH2 (Fig. 6H, I and Supplementary Fig. 6I)" they missed out on referring to Fig. 6J here, and further down in lines 336 they incorrectly reference the ChIP data in Figure 6 (for example, they reference the histone acetylation ChIP data as being in panels K and L, whereas it is L and M).

2) The pLKO CI-994 curve is a bit too steep for accurately calculating an IC₅₀/EC₅₀, but I don't consider this to be a major point as there is a clear shift in the curve. Even so, I think this needs to be acknowledged.

Point by point response to Reviewers' comments:

Reviewer #2 (Remarks to the Author):

The authors have addressed most of the raised concerns.

We thank the reviewer for the helpful suggestions and appreciating the revision made.

Reviewer #3 (Remarks to the Author):

In the present revised manuscript, the authors found that increased expression of the ubiquitin ligase RNF5 contributes to AML development.

The manuscript might have prognostic and therapeutic applications.

Given that the over-expression of RNF5 in cancer was previously reported Cancer Res 2007 Sep 1;67(17):8172-9. doi: 10.1158/0008-5472.CAN-07-0045 this study originally identifies the axis RNF5/RBBP4 and its potential as prognostic role.

In this revised version the authors have improved the evidences in support of their hypothesis.

Despite some points stay unanswered (as an example i) RNF5-RBBP4 in HDACi-resistant unresponsive cells;ii) global genomic assessment of RBBP4-dependent changes) the medical potential of the study is clear and so are the following applications.

We thank the reviewer for helpful comments and for appreciating the therapeutic implications of our work. While we recognize the importance of global genomic assessment of RBBP4-dependent changes, we believe this type of analysis is a starting point to a future study designed to identify pathways regulated by RNF5-modified RBBP4.

Reviewer #4 (Remarks to the Author):

In this paper, the authors identify high protein levels of the E3 ligase RNF5 as being associated with a poor prognosis in AML. They go on to show that RNF5 knockdowns reduce leukaemia growth both in vitro and in vivo. Importantly, they show that wild type RNF5 expression can rescue the knockdowns while an enzymatic mutant of RNF5 fails to rescue the growth defect. Using an MLL-AF9 retroviral transduction model, they also show that transduced cells from Rnf5^{-/-} mice produce a leukaemia with a longer latency in an engraftment experiment. They also use the LINCS database to identify an overlapping gene expression signature between cells treated with HDAC inhibitors and those that harbour an RNF5 knockdown. A mass spec experiment allowed them to identify RNF5 interacting proteins, and they performed a thorough characterization of one factor, RBBP4. The main result from this section was that RNF5 mediated ubiquitination of RBBP4 had no impact on stability or localization in the cell, but did alter chromatin binding and loss of RNF5 protein levels caused increased histone acetylation at a defined subset of RBBP4 gene targets. Finally, in the last section of the paper, using an inhibitor screen they show that HDAC inhibitors synergize with RNF5 knockdowns, and they characterize one HDAC inhibitor in some detail. The overall conclusion is that low RNF5 protein levels could be a general predictor of HDAC inhibitor sensitivity.

I did not review the first version of this paper, but overall I think this is quite a comprehensive and

interesting study. I read over the reviewer's comments, and in my opinion the authors have addressed many of the main points raised in previous reviews. I just have two minor points below.

We thank the reviewer for appreciating the significance of our study and for helpful comments.

Minor points

1) Line 334, "HDAC2, or EZH2 (Fig. 6H, I and Supplementary Fig. 6I)" they missed out on referring to Fig. 6J here, and further down in lines 336 they incorrectly reference the ChIP data in Figure 6 (for example, they reference the histone acetylation ChIP data as being in panels K and L, whereas it is L and M).

We thank the reviewer for noting this. This has now been corrected.

2) The pLKO CI-994 curve is a bit too steep for accurately calculating an IC50/EC50, but I don't consider this to be a major point as there is a clear shift in the curve. Even so, I think this needs to be acknowledged.

We thank the reviewer for allowing us to clarify these points. We agree with the reviewer that pLKO response curve for CI-994 is too steep for accurately calculating the IC50 since control cells didn't respond well to the drug under the concentration and time point used. Therefore, we didn't calculate CI-994 IC50 in these graphs but have shown that there is a clear and significant difference in the response of AML cells to the drug following RNF5 KD (as shown in Fig. 7c and Supplementary Fig. 7a).